# Aminoacyl chain translocation catalysed by a type II thioesterase domain in an unusual non-ribosomal peptide synthetase

Shan Wang [1,7], William D. G. Brittain [2,7], Qian Zhang[3,7], Zhou Lu[1], Ming Him Tong[1], Kewen Wu[1], Kwaku Kyeremeh [4], Matthew Jenner [5,6✉], Yi Yu [3✉], Steven L. Cobb [2✉] & Hai Deng [1✉]

Non-Ribosomal Peptide Synthetases (NRPSs) assemble a diverse range of natural products with important applications in both medicine and agriculture. They consist of several multienzyme subunits that must interact with each other in a highly controlled manner to facilitate efficient chain transfer, thus ensuring biosynthetic fidelity. Several mechanisms for chain transfer are known for NRPSs, promoting structural diversity. Herein, we report the first biochemically characterized example of a type II thioesterase (TE_{II}) domain capable of catalysing aminoacyl chain transfer between thiolation (T) domains on two separate NRPS subunits responsible for installation of a dehydrobutyrine moiety. Biochemical dissection of this process reveals the central role of the TE_{II}-catalysed chain translocation event and expands the enzymatic scope of TE_{II} domains beyond canonical (amino)acyl chain hydrolysis. The apparent co-evolution of the TE_{II} domain with the NRPS subunits highlights a unique feature of this enzymatic cassette, which will undoubtedly find utility in biosynthetic engineering efforts.

[1] Department of Chemistry, University of Aberdeen, Aberdeen AB24 3UE, UK. [2] Department of Chemistry, Durham University, Science Site, Durham DH1 3LE, UK. [3] Key Laboratory of Combinatorial Biosynthesis and Drug Discovery (MOE) and Hubei Province Engineering and Technology Research Centre for Fluorinated Pharmaceuticals, School of Pharmaceutical Sciences, Wuhan University, Wuhan 430071, China. [4] Department of Chemistry, University of Ghana, P.O. Box LG56 Legon-Accra, Ghana. [5] Department of Chemistry, University of Warwick, Coventry CV4 7AL, UK. [6] Warwick Integrative Synthetic Biology (WISB) Centre, University of Warwick, Coventry CV4 7AL, UK. [7] These authors contributed equally: Shan Wang, William D. G. Brittain and Qian Zhang. ✉email: M.Jenner@warwick.ac.uk; yu_yi@whu.edu.cn; s.l.cobb@durham.ac.uk; h.deng@abdn.ac.uk

Nonribosomal peptides (NRPs) are a structurally diverse class of secondary metabolites produced by large multimodular enzymes known as nonribosomal peptide synthetases (NRPSs)[1]. During biosynthesis, NRPSs utilize a modular thio-template mechanism, typically consisting of multidomain enzymatic subunits that must interact with each other in a tightly controlled manner during chain translocation events to ensure biosynthetic fidelity[2]. The reactions catalysed by catalytic domains within each module fulfil a cycle of peptidyl chain elongation using at least three domains; adenylation (A), condensation (C), and thiolation (T)[2]. The A domains selectively activate and load amino acid extender units onto a 4'-phosphopantetheine (Ppant) moiety, which is post-translationally appended to each T domain in the corresponding module, and the C domains typically catalyse chain extension reactions between the growing peptidyl chain and the downstream amino acid. During the biosynthetic process, the growing peptide intermediates remain covalently tethered to the T domains via a thioester linkage. Chain release of the nascent peptide chain is then achieved by either hydrolysis or intramolecular cyclization to generate either a linear or cyclic peptide, respectively[2].

While aminoacyl/peptidyl chain transfer events are commonly mediated by C domains, whereby the upstream T-bound intermediate unit is transferred to the downstream T domain through the formation of a peptide bond (Fig. 1a)[2], Nature has evolved several other mechanisms of chain translocation to downstream pathways. For example, ketosynthase (KS)-mediated chain transfer of T-tethered intermediates are often found in NRPS-PKS/fatty acid hybrid pathways. This is exemplified in the biosynthesis of pyrrolomycin C **1**, where the first KS domain of Pyr25, a multidomain PKS complex, is proposed to catalyse aminoacyltransfer between a Pyr28-tethered chlorinated pyrrole intermediate and the downstream acyl carrier protein (ACP)-bound acyl unit (Fig. 1b)[3]. An alternative mechanism is to use a $TE_{II}$ enzyme to hydrolyse the T-tethered substrates which creates a pool of a given specialised amino acid dedicated for an A domain in the downstream NRPS. An example of this phenomenon can be found in the biosynthesis of barbamide.[4,5] Here, the didomain NRPS subunit, BarA, activates L-Leu, and a T domain-tethered Leu residue is then subjected to three chlorination events at the unreactive methyl group, followed by transamination to generate BarA-tethered α-ketotrichloroisocaproic acid. Subsequent hydrolysis catalysed by the $TE_{II}$, BarC, yields the corresponding chlorinated α-ketoacid which will be loaded onto the main NRPS assembly line, BarE (Fig.1c).

A further example is acyltransferase (AT) domain-mediated transfer, where AT domains catalyse the condensation between a T domain-tethered aminoacyl unit and a free-standing substrate. This particular approach can be found in the biosynthesis of coumermycin $A_1$ where CouN7 transfers a pyrrole acid unit to ultimately assemble the pyrrole-based product[6]. A rather unusual case of AT domain-catalysed chain translocation was found in the biosynthesis of coronamic acid **3** where an AT-like domain, CmaE, shuttles the substrate between T domains of different NRPS enzymes (Fig. 1d)[7]. The didomain NRPS, CmaA, activates isoleucine, followed by aminoacyltransfer from CmaA to a standalone T domain, CmaD, catalysed by CmaE. CmaD delivers the substrate to the subsequent enzyme for the conversion to **3**. Such CmaE-mediated aminoacyltransfer is essential for the biosynthesis, as the downstream enzymes cannot accept CmaA-tethered substrates. CmaE-like AT homologues are also found in the biosynthetic pathways of syringomycin E[8] and zorbamycin[9]. Another example involves spontaneous transthiolation between T domains which was observed in pathway reconstitution of the antibiotic natural product, pacidamycin S **4** (Fig. 1e)[10]. PacP is an A-T-$TE^0$ tridomain NRPS, of which the $TE^0$ domain was believed

to be inactive due to lack of the typical catalytic triad of TE domains. Incubation of PacP-tethered substrate with PacH, a standalone T domain, resulted in high transfer efficiency of an aminoacyl group from the T domain of PacP to PacH, indicating that the transthiolation occurs spontaneously[10].

Legonmycins **5** (Fig. 1f) belong to a group of bacterial pyrrolizidine alkaloids (PAs)[11]. Compared to other bacterial PAs, such as pyrrolizixenamides[12], azabicyclene[13,14], and bohemamine derivatives[15–18] (Supplementary Fig. 1a), legonmycins contain an extra methyl group at the C7 position (Fig. 1f)[19]. Unlike the biosynthetic pathways of other bacterial PAs, which have canonical NRPS complexes, the pathway for the assembly of legonmycins is rather unusual. It contains two NRPS proteins, LgnB and LgnD, with domain arrangement of $A_1$-$T_0$ and $C_1$-$T_1$-$C_2$-$A_2$-$T_2$-TE, respectively[19]. The $A_1$ domain in LgnB and $A_2$ in LgnD were predicted to activate L-Thr and L-Pro, respectively (Supplementary Fig. 1b). It was proposed that the biosynthesis of legonmycins **5** is initiated by the activity of LgnB and LgnD, utilising L-Thr, L-Pro, and acyl-CoAs to generate the final NRPS-bound acylated-dehydrobutyrin(Dhb)-Pro intermediate, which is likely followed by a TE-catalysed ring closure, leading to the key intermediates, legonindolizidines **6** (Supplementary Fig. 1b). Biochemical analyses showed that the monooxygenase, LgnC, acts on **6** and catalyses a multistep reaction to yield **5** (Supplementary Fig. 1b)[19]. In contrast to the incorporation of Ser in other PA systems[12–18], the usage of a L-Thr residue in the biosynthesis of **5** was believed to contribute the extra methyl group at C7 of **5**[19]. Another unique feature is that the minimal biosynthetic gene cluster (BGC) (accession number: KM514925) of **5** contains an essential gene, *lgnA*, encoding a type II thioesterase ($TE_{II}$) orthologue, which was proposed to play an important editing role of removing as-yet-known aberrant intermediates[19]. Interestingly, two amide bonds are formed in the proposed biosynthesis of **6**, consistent with two C domains in LgnD (Supplementary Fig. 1b), while three T domains were found in LgnB and LgnD[19]. This raises an interesting question of whether a chain translocation event between T domains is required.

Herein we report an extensive set of biochemical experiments that establish the role played by LgnA, LgnB-$T_0$ and LgnD-$T_1$ domains in the unusual chain translocation employed at the early stage of legonindolizidine NRPS (Fig. 1f). These experiments demonstrate that the $TE_{II}$ orthologue, LgnA, transfers a L-Thr unit from the LgnB-$T_0$ to the LgnD-$T_1$ domain, allowing condensation with isovaleryl (IV)-CoA to yield a $T_1$-bound IV-Thr unit. This is followed by a LgnD-$C_2$ domain-catalysed dehydration to provide the key intermediate, $T_1$-bound IV-Dhb for the downstream NRPS. Importantly, LgnA also efficiently hydrolyses $T_0$-tethered IV-Thr units that form via aberrant activity of the LgnD-$C_1$ domain, thus maintaining biosynthetic fidelity. Taken together, these results demonstrate that the recruitment of a specialised class of "adapter" $TE_{II}$ proteins to biosynthetic clusters has likely enabled evolutionary distinct NRPS subunits to communicate via an aminoacyltransfer reaction, thereby increasing the diversity of natural product structure while mitigating the risk of biosynthetic derailment.

## Results

**LgnA $TE_{II}$ is essential for production of legonindolizidin A 6.**
To investigate the role of the LgnB and LgnD NRPS subunits in the production of **6**, we first sought to overexpress both *lgnB* and *lgnD* genes in *E. coli* BL-21 CodonPlus (DE3)-RP. LgnB was expressed as an N terminal $pHis_6$-recombinant protein while LgnD was a C-terminal $pHis_6$-recombinant protein, and both were produced in a soluble form that was purified to near homogeneity by Ni-NTA chromatography and gave estimated

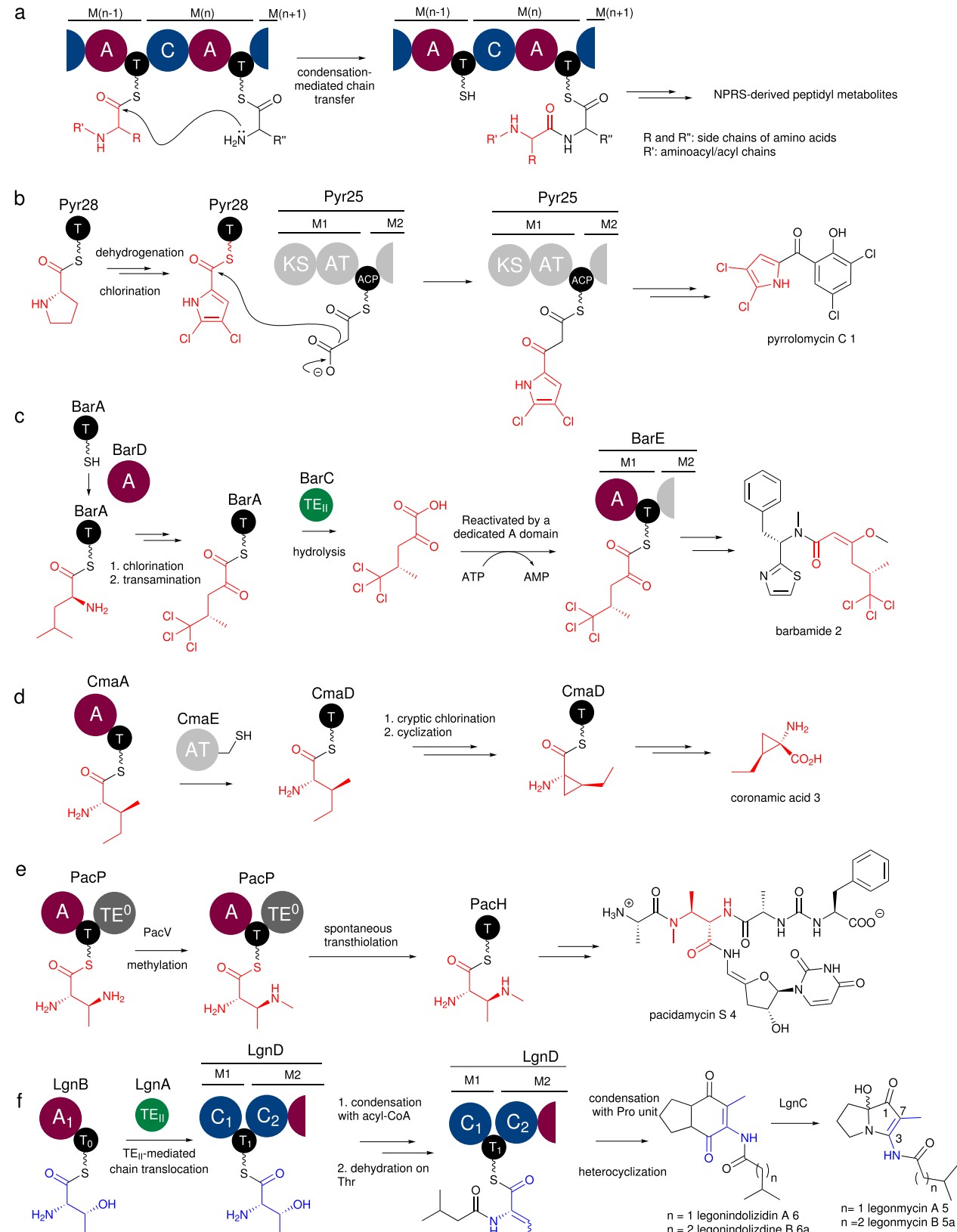

molecular weights of 65 kDa for LgnB and 207 kDa for LgnD as observed in SDS page analysis (Supplementary Fig. 2).

In the first instance, *apo*-LgnB and LgnD were converted into their *holo* forms, using the Ppant transferase, Sfp, in the presence of coenzyme A and $MgCl_2$. This was followed by the addition of substrates, L-Thr, L-Pro, IV-CoA, and ATP to initiate the biosynthetic process. The assay was monitored by HPLC and UHPLC-high resolution-electrospray ionisation-mass spectrometry (HR-MS). However, no desired product was detected (Fig. 2a (i) and Supplementary Fig. 3).

Given the essential role of the *lgn*A gene during the biosynthesis of **6** from our previous observations[19], we then

**Fig. 1 Mechanisms of aminoacyl chain translocation to downstream thiotemplated pathways. a** Intrasubunit chain condensation/translocation catalysed by a C domain in type I NRPSs. **b** Ketosynthase (KS) domain-mediated chain translocation in the hybrid PKS-NRPS responsible for the biosynthesis of pyrrolomycin C **1**. **c** Chain translocation via $TE_{II}$ domain-mediated acyl-thioester hydrolysis followed by reactivation by a dedicated A domain during the biosynthesis of barbamide **2**. **d** AT domain-mediated aminoacyltransfer between T domains in the biosynthesis of coronamic acid **3**. **e** Spontaneous transthiolation between PacP and PacH has been observed in the in vitro pathway reconstitution of pacidamycin S **4**. **f** A new aminoacyl chain translocation occurring between T domains catalysed by a $TE_{II}$ orthologue in the production of legonindolizidines **6**. A, adenylation domain (purple); C, condensation domain (blue); T, thiolation/peptidyl carrier protein domain (black), TE, thioesterase domain (dark grey) or standalone $TE_{II}$s (green), $TE^0$: inactive TE domain. All polyketide domains are shown in light grey and are abbreviated as follows: AT, acyltransferase domain; KS, ketosynthase domain; ACP, acyl carrier protein. The chain moieties that are transferred are highlighted in red while the Thr residue in the biosynthesis of legonindolizidines **6** is coloured in blue.

overexpressed LgnA in *E. coli* as a N-terminal $His_6$-recombinant protein, followed by purification to near homogeneity with the expected molecular weight (29 kDa) (Supplementary Fig. 4). Subsequently, an assay comprised of LgnA (0.5 µM), LgnB (10 µM) and LgnD (10 µM), together with L-Thr, L-Pro, IV-CoA, and ATP was performed. HPLC analysis clearly demonstrated that **6** was produced, compared to an authentic standard (Fig. 2a (**ii**) and (**iii**)), indicating that the presence of LgnA facilitates the production of **6**.

In order to investigate T-bound intermediates, we applied a chemical derivatization strategy to offload the biosynthetic intermediates attached to the NRPS subunits. This approach uses an excess of cysteamine to cleave the (amino)acyl-thioesters attached to the Ppant arm of each T domain. This transthioesterification reaction, followed by intramolecular rearrangement and disulphide formation with another free cysteamine unit, yields a free amine that is readily detectable by positive mode HR-MS[20,21]. Based on previous studies[12–19], the chemically captured derivatives of possible biosynthetic intermediates for **6** were predicted to be the cystamine adducts of Thr **7**, IV-Thr **8**, IV-Dhb **9**, IV-Thr-Pro **10**, and IV-Dhb-Pro **11** as shown in Fig. 2b.

To this end, *holo* LgnB was first incubated with L-Thr and ATP in the presence of cysteamine. HR-MS analysis of the enzymatic mixture produced an ion corresponding to **7**, indicating that the LgnB $A_1$ domain indeed activates L-Thr which is subsequently loaded onto the LgnB-$T_0$ domain (Supplementary Fig. 5). An assay of *holo*-LgnB and *holo*-LgnD together with substrates in the presence of cysteamine was carried out, which afforded the detection of new ions corresponding to **7**, **8**, and **10**, however, derivatives, **9** and **11**, were not observed (Supplementary Fig. 6). To confirm the identity of the cystamine adducts, we synthesized IV-Thr-Pro-SNAC **10a** (Supplementary Fig. 7–9) as a synthetic mimic of protein-tethered IV-Thr-Pro. Incubation of **10a** in the presence of cysteamine allowed the generation of **10**, confirming the occurrence of chemical cleavage and rearrangement of the thioesters (Supplementary Fig. 10). When LgnA (0.5 µM) was introduced to the above assay, HR-MS analysis confirmed the presence of ions corresponding to derivatives **7–11**, indicating that LgnA facilitates the Dhb formation (Fig. 2c and Supplementary Fig. 11). Removal of L-Pro from this assay led to the accumulation of only **7**, **8**, and **9** (Supplementary Fig. 12). The presence of **8** and **9** in this assay suggested that the dehydration occurs on the NRPS-tethered IV-Thr intermediate to yield the IV-Dhb intermediate, prior to the condensation with L-Pro to generate LgnD-tethered IV-Dhb-Pro.

The captured IV-Thr-Pro intermediate **10** is likely a dead-end species, arising from the condensation between an L-Pro unit and NRPS-tethered IV-Thr (presumably $T_0$-tethered IV-Thr, *vide infra*) that escapes the dehydration. Interestingly, **10** was found to accumulate in both assays in the presence and absence of LgnA. Our time-course experiments showed that a significantly larger amount of **10** was accumulated in the assay when LgnA was omitted over a period of 3 h, compared to when LgnA was

present (Supplementary Fig. 13). A possible explanation for this observation is that, even when LgnA is present, the nonselective condensation domain (presumably LgnD-$C_2$ domain) is still able to condense the NRPS-tethered IV-Thr unit with the downstream L-Pro unit to produce NRPS-tethered IV-Thr-Pro.

**LgnA facilitates the Dhb formation catalysed by LgnD-$C_2$ domain on LgnD-$T_1$ domain.** To understand how LgnA facilitates Dhb formation, it was necessary to investigate the exact location of the dehydration in the NRPS assembly. Our phylogenetic analysis suggested that the LgnD-$C_2$ domain belongs to a group of unusual $C^*$ domains (Supplementary Fig. 14), analogous to the recently characterized AlbB-$C_2$ and -$C_3$ domains in the biosynthetic pathway of albopeptide, which catalyse the dehydration on L-Ser and L-Thr to dehydroalanine (Dha) and (*E*)-Dhb residues, respectively[21]. Similar to AlbB-$C_2$ and -$C_3$, LgnD-$C_2$ possesses a HHxxDG catalytic motif, characteristic of conventional C domains, where the second $\underline{H}$ is proposed to catalyse the amide formation. This is distinct from the NocB-$C_5$ domain, in nocardicin biosynthesis, which contains a rather unique $\underline{H}_{(790)}$HHxxDG motif in its active site where $H_{790}$ was shown to be responsible for the generation of a transient Dha species, while $H_{792}$ catalyses amide bond formation during the course of the β-lactam formation[22,23]. However, structural information is needed to determine the residue(s) responsible for the dehydration in these $C^*$ domains including LgnD-$C_2$.

We chose to overexpress two truncated proteins; LgnD-$C_1$-$T_1$ didomain and LgnD-$C_1$-$T_1$-$C_2$ tridomain in *E. coli*, followed by purification to near homogeneity with the expected molecular weights of 66 kDa and 114 kDa, respectively, as observed in SDS page analysis (Supplementary Fig. 15). Compared to the earlier assay containing LgnA, LgnB and the full-length LgnD where both **8** and **9** were present (Fig. 2d (**i**)), incubation of LgnB and LgnD-$C_1$-$T_1$ in the presence or absence of LgnA confirmed the presence of one ion congruent with the captured intermediate **8** as indicated by our HR-MS analysis (Supplementary Fig. 16), suggesting that LgnA is not required for the formation of the IV-Thr unit. However, the absence of **9** (Fig. 2d (**ii**)) indicated that the dehydration reaction on the IV-Thr unit had not occurred in this assay. We also mutated the key Ser residue (S492) of LgnD-$T_1$ in the truncated LgnD-$C_1$-$T_1$ didomain to Ala to yield the LgnD-$C_1$-$T_1$(S492A) variant, preventing the *apo*-LgnD-$T_1$ domain from being converted to the *holo* form. HR-MS analysis of the assay of LgnA, LgnB and LgnD-$C_1$-$T_1$(S492A) showed the occurrence of only **8** (Supplementary Fig. 16), indicating that IV-Thr must be bound to the LgnB-$T_0$ domain in the case with the LgnD-$C_1$-$T_1$(S492A) variant.

Incubation of LgnD-$C_1$-$T_1$-$C_2$ with LgnA and LgnB in the presence of cysteamine allowed the accumulation of both **8** (Supplementary Fig. 17) and **9** as observed in our HR-MS analysis (Fig. 2d (**iii**)), confirming that the presence of the $C_2$ domain is a prerequisite for the dehydration reaction. To ascertain which T domain the dehydration reaction is occurring on, we generated a

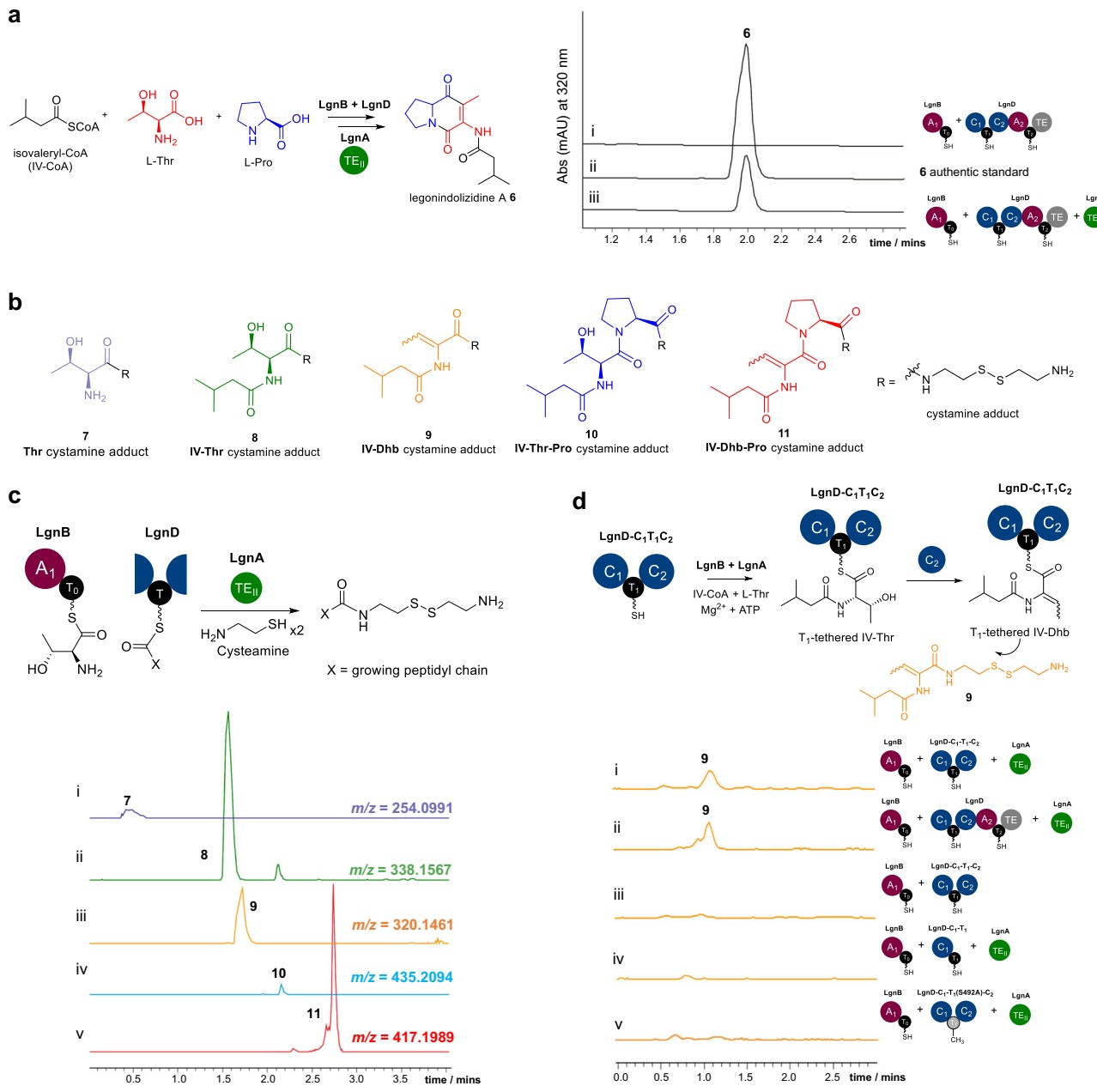

**Fig. 2 HPLC and chemical capturing-based LC-MS analyses of the production of 6 and potential biosynthetic intermediates, 7–11, respectively, in the assays of the pathway reconstitution and truncated LgnD with LgnB in the presence or absence of LgnA. a** HPLC analysis of the production of **6** in the in vitro reconstitution experiments, (i) the assay of LgnB and LgnD only, (ii): the authentic **6**; (iii), the assay of LgnA, LgnB and LgnD. **b** The structures of the expected cystamine-adducted intermediates, **7–11**. **c** LC-MS traces of chemically captured intermediates during the NRPS assembly towards **6** in the presence of LgnA. **d**. LC-HR-MS analysis of chemical captured intermediate, **9**, in the assays of truncated LgnD constructs, (i). the extracted ion chromatograms (EICs) of **9** in the assay of LgnB with truncated LgnD-$C_1$-$T_1$-$C_2$; (ii); the EICs of **9** in the assay of LgnB and truncated LgnD-$C_1$-$T_1$-$C_2$ in the presence of LgnA; (iii). the EICs of **9** in the assay of LgnB and truncated LgnD-$C_1$-$T_1$ in the presence of LgnA. (iv). the EICs of **9** in the control by incubation of LgnB, LgnD and LgnA. (v). the EIC of 0 in the assay of LgnA, LgnB and the truncated LgnD-$C_1$-$T_1$(S492A)-$C_2$ tridomain. The molar ratio of LgnA to LgnB and LgnD or truncated LgnD was 1:20:20. The individual intermediate derivative and the corresponding LC-MS traces were colour coded. All LC-MS traces were represented in the same scale of ion intensity ($10^4$).

LgnD-$C_1$-$T_1$(S492A)-$C_2$ variant. In an assay using this variant with LgnA, only intermediate **8** (Supplementary Fig. 17) was observed, but not **9** (Fig. 2d (**iv**)). This indicated that the presence of a *holo*-LgnD-$T_1$ domain is key for the dehydration reaction on the IV-Thr unit to provide IV-Dhb species. Taken together, our results strongly support that the LgnD-$C_2$ domain catalyses the dehydration reaction of LgnD-$T_1$-tethered IV-Thr to yield an IV-Dhb unit.

Interestingly, when omitting LgnA in an assay of LgnB-$A_1$-$T_0$ and LgnD-$C_1$-$T_1$-$C_2$, only intermediate **8** was observed (Supplementary Fig. 17) but not **9** (Fig. 2d (**v**)), demonstrating that IV-Thr unit is likely tethered to LgnB-$T_0$ and not the LgnD-$T_1$ domain, as a LgnD-$T_1$-bound IV-Thr unit would undergo a dehydration reaction in the presence of LgnD-$C_2$ domain. This observation raised the question of whether LgnA transfers Thr or

IV-Thr or indeed both units, from LgnB-$T_0$ to LgnD-$T_1$ domain for the dehydration reaction.

**LgnA catalyses the chain translocation of $T_0$-tethered Thr unit to LgnD-$T_1$ domain.** Intact protein mass spectrometry (MS) is an invaluable tool for the interrogation of covalently tethered intermediates of polyketide synthases (PKS) and NRPSs[24–26]. To this end, we adopted a stepwise approach to accumulate T domain-tethered intermediates in the early stages of the biosynthesis of **6** that can then be observed by intact protein MS. Taking the *holo* form of LgnB $A_1$-$T_0$ (Fig. 3b (**i**)) as the starting point, following addition of L-Thr and ATP we observed a new species from the deconvoluted mass spectrum (Fig. 3b (**ii**)). This new species corresponded to the addition of one L-Thr unit tethered to LgnB-$A_1$-$T_0$, as indicated by the mass shift of +101 Da relative to the mass of *holo*-LgnB $A_1$-$T_0$.

Next, we performed an assay using LgnB-$A_1$-$T_0$ (100 μM) and LgnD-$C_1$-$T_1$ (100 μM) and L-Thr, in the absence and presence of LgnA. Compared to the mass of *holo*-LgnD-$C_1$-$T_1$ (Fig. 3b (**iii**)), only LgnB-$A_1$-$T_0$-tethered L-Thr was observed in the absence of LgnA (Fig. 3b (**iv**)), with no species corresponding to LgnD-$C_1$-$T_1$ tethered L-Thr observed. In contrast, when LgnA (5 μM) was included, a +101 Da mass shift relative to *holo*-LgnD-$C_1$-$T_1$ was clearly observed for a new species corresponding to one unit of L-Thr tethered to LgnD-$C_1$-$T_1$ (Fig. 3b (**v**)), strongly indicating that LgnA catalyses an aminoacyltransfer of L-Thr units from the LgnB-$T_0$ to LgnD-$T_1$ domain. It should be noted that there is a population of LgnB-$A_1$-$T_0$-tethered L-Thr present due to the excess L-Thr added in the enzyme mixture (Fig. 3b (**v**)). An attempt to capture the possible acylated LgnA intermediate during this aminoacyltransfer was carried out. To this end, we incubated *holo*-LgnB-$A_1$-$T_0$ (100 μM) and LgnA (100 μM) in the presence of L-Thr and ATP and monitored LgnA by intact protein MS. Compared to the control of LgnA alone (Fig. 3c (**i**)), incubation with *holo*-LgnB-$A_1$-$T_0$, L-Thr and ATP allowed almost all of LgnA to become aminoacylated with a single L-Thr unit (+ 101 Da) (Fig. 3c (**ii**)). Observation of a stable aminoacyl-enzyme intermediate suggested that LgnA does not hydrolyse L-Thr from the LgnB-$T_0$ domain, but acts as an aminoacyltransferase to shuttle L-Thr between NRPS subunits.

LgnA possesses a canonical catalytic triad ($Ser_{77}$-$Asp_{184}$-$His_{212}$) exhibited by classical $TE_{II}$ domains with the active site Ser residue situated in a conserved $G_{75}$-$H_{76}$-**$\underline{S}_{77}$**-$X_{78}$-$G_{79}$ motif. These features have been observed in well-characterised hydrolytic $TE_{II}$ domains, such as RifR[27] in the polyketide pathway of rifamycin and BorB[28] in the PKS-NRPS pathway of borrelidin (Supplementary Fig. 18). It is worth noting that $TE_{II}$ domains belong to the α,β-hydrolase superfamily, which also includes AT domains from polyketide synthases, where the Ser residue in a Ser-His dyad is acylated by an acyl-CoA species before being transferred to the Ppant thiol of the carrier protein domain[29]. It is therefore likely that the equivalent Ser residue ($Ser_{77}$) in LgnA is the active site residue that becomes acylated with L-Thr. The $Ser_{77}$ residue of LgnA was then mutated to Ala in order to yield the inactive variant, LgnA(S77A). Addition of LgnA(S77A) in the pathway reconstitution assays abolished the production of **6** (Supplementary Fig. 19a) when compared to the authentic standard of **6** and the control reaction, as observed in HPLC and LC-MS analysis (Supplementary Fig. 19a and b). Chemical capturing in these reconstitution experiments also indicated the absence of key intermediate cystamine adducts, **9** and **11**, in the reaction with LgnA(S77A), compared to the control (Supplementary Fig. 19c and d). These snapshots of our intact MS analysis together with site directed mutagenesis (SDM) and chemical capturing experiments strongly demonstrated that LgnA

transfers an L-Thr residue from LgnB-$T_0$ domain to LgnD-$T_1$ domain via an L-Thr aminoacyl LgnA intermediate.

**LgnA hydrolyses the aberrant $T_0$-tethered IV-Thr units.** While we had managed to elucidate the aminoacyltransferase role of LgnA, the canonical hydrolytic activity of $TE_{II}$ domains had not been probed. To investigate this further, a pre-incubation of *holo*-LgnB $A_1$-$T_0$ (100 μM), *holo*-LgnD-$C_1$-$T_1$ (100 μM), L-Thr, ATP and LgnA (5 μM) for 30 mins allowed the $T_0$ and $T_1$ domains to be fully acylated with a L-Thr unit (Supplementary Fig. 20iii). Addition of IV-CoA into the reaction mixture resulted in observation of an IV-Thr unit attached to LgnD-$C_1$-$T_1$ as the stalled intermediate due to the absence of the downstream NRPS machinery (Supplementary Fig. 20iv), consistent with our chemical capturing results that show IV-Thr is correctly formed on LgnD-$T_1$ domain. Interestingly, in the same assay, IV-Thr is also found tethered to LgnB-$A_1$-$T_0$. This is probably due to full occupancy of the LgnD-$T_1$ domain with IV-Thr units, leaving the nonselective $C_1$ domain to catalyse the formation of IV-Thr on LgnB-$T_0$ domain. Again, this is in agreement with our previous chemical capturing results in the assay between LgnB and LgnD-$C_1$-$T_1$ didomain (Supplementary Fig. 16). The appearance of *holo*-LgnB was interesting as LgnB should be fully occupied when excessive substrates were added, suggesting aminoacyl chain hydrolysis (Supplementary Fig. 20iv). This was confirmed upon observation that the hydrolytic product, IV-Thr-COOH **12**, was accumulated in the pathway reconstitution assay with LgnA as indicated in HR-MS and $MS^2$ fragmentation analyses, while no **12** was observed in the assay without LgnA (Supplementary Fig. 21). This clearly indicates hydrolytic activity of LgnA towards the IV-Thr unit.

To probe this in more detail, a chemoenzymatic synthesis of IV-Thr-CoA was attempted[30], which could be used for artificial loading onto T-domains and assessing hydrolytic activity of LgnA. While the construction of IV-*O*-*t*Bu-Thr-COOH **12a** and pantetheine arm (pant) **13** were straightforward (Supplementary Methods and Supplementary Fig. 22). Coupling **12a** with **13** proved to be challenging, resulting in unsuccessful isolation of the desired products by either flash or HPLC chromatography. Instead, IV-*O*-*t*Bu-Thr-SNAC **14** was synthesized (Supplementary Fig. 23), which could be deprotected under acidic conditions to yield IV-Thr-SNAC **14a**. Interestingly, two conformational isomers of **14a** appeared in our LC-MS analysis (Supplementary Fig. 24), suggesting the presence of a possible equilibrium between *cis*- and *trans*-configurations of the amide bond. Incubation of **14a** with LgnA or LgnA(S77A), however, showed that no hydrolytic reaction occurred (Supplementary Fig. 24).

Next, we pursued the in situ production of the IV-Thr unit on the LgnB-$T_0$ domain to check the hydrolytic activity of LgnA. Here, LgnB (100 μM) and LgnD-$C_1$-$T_1$ (100 μM) were co-incubated with L-Thr, IV-CoA and ATP to generate an IV-Thr unit tethered to LgnB-$T_0$ domain (Fig. 4d (**iii**)) (no IV-Thr formation on LgnD-$C_1$-$T_1$ as no LgnA was present to catalyse transfer). In order to prevent further loading/condensation events, the excess L-Thr, IV-CoA and ATP were then removed from the assay by ultrafiltration. The addition of LgnA (5 μM) into the mixture resulted in formation of *holo*-LgnB-$A_1$-$T_0$ (Fig. 4d (**iv**)), unambiguously confirming the hydrolytic activity of LgnA. The $T_0$-tethered IV-Thr remained intact when the LgnA(S77A) variant was used in the equivalent assay (Fig. 4d (**v**)), confirming that the $S_{77}$ residue in LgnA is crucial for the hydrolytic reaction. Taken together, the intact MS analyses and SDM strongly supports that LgnA hydrolyses LgnB-$T_0$-tethered IV-Thr to **12** and also that LgnA is incapable of transferring the IV-Thr unit from LgnB-$T_0$ to LgnD-$T_1$ domain.

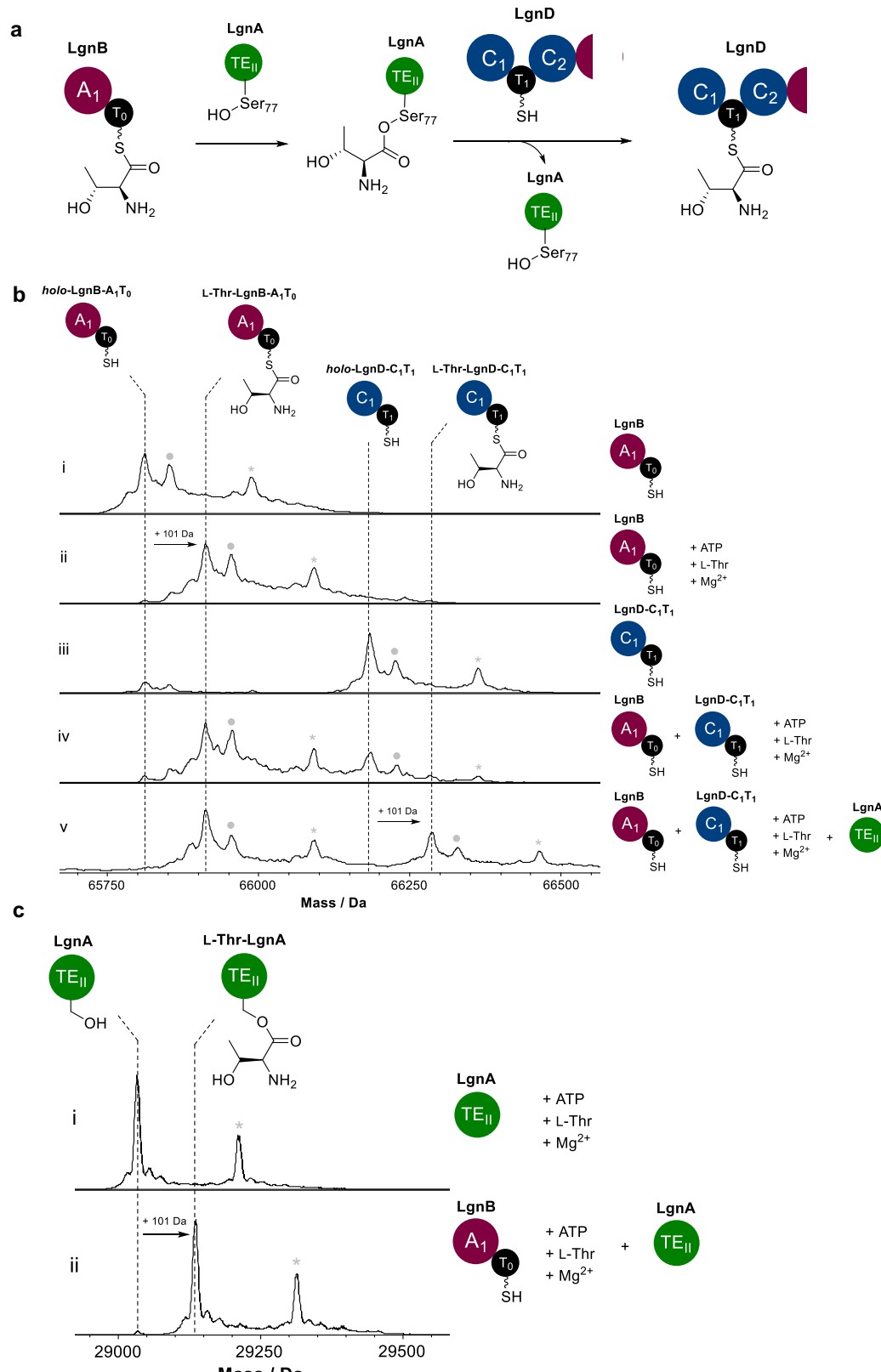

Interestingly, the concentration of LgnA in the enzyme mixture is crucial for the production of **6**. The optimal molar ratio of LgnA to LgnB and LgnD is approximately 1:20, where the production of **6** is the most efficient (Supplementary Fig. 25a) and the least amount of the hydrolytic product, IV-Thr-COOH, **12**, was observed (Supplementary Fig. 25b). Increasing the LgnA ratio

in the enzyme mixture resulted in decreased production of **6** but the increased accumulation of **12** (Supplementary Fig. 25).

When the ratio of LgnA to LgnB and LgnD-$C_1$-$T_1$ was 1:4, no IV-Thr was observed attached to either LgnB $A_1$-$T_0$ or LgnD-$C_1$-$T_1$ in our intact MS analysis, with only a Thr unit observed attached to LgnD-$C_1$-$T_1$ (Supplementary Fig. 26), suggesting that

**Fig. 3 LgnA catalyses the aminoacyltransfer of Thr unit from LgnB-$T_0$ to LgnD-$T_1$ domains. a** A schematic diagram of the aminoacyl translocation catalysed by LgnA. **b** Deconvoluted intact protein mass spectra of (i) *holo*-LgnB $A_1$-$T_0$, (ii) *holo*-LgnB-$A_1$- $T_0$ + ATP + L-Thr + $Mg^{2+}$, (iii) *holo*-LgnD-$C_1$- $T_1$, (iv) *holo*-LgnB $A_1$-$T_0$ + *holo*-LgnD-$C_1$- $T_1$ + ATP + L-Thr + $Mg^{2+}$, (v) *holo*-LgnB $A_1$-$T_0$ + *holo*-LgnD-$C_1$- $T_1$ + ATP + l-Thr + $Mg^{2+}$ + LgnA-$TE_{II}$. **c** Deconvoluted intact protein mass spectra of (i) LgnA-$TE_{II}$ + ATP + L-Thr + $Mg^{2+}$, (ii) LgnA-$TE_{II}$ + *holo*-LgnB-$A_1$-$T_0$ + ATP + L-Thr + $Mg^{2+}$. Peaks labelled with grey dots and asterisks in the intact MS spectra indicate N-terminal acetylation or gluconoylation, respectively, both of which are known post-translational modifications of recombinant heterologous proteins in *E. coli*[24]. The exact measured and observed masses for each species are detailed in Supplementary Table 3.

the stalled IV-Thr on LgnD-$C_1$-$T_1$ and the aberrant IV-Thr on LgnB-$A_1$-$T_0$ were hydrolysed by the excessive amount of LgnA. This is consistent with the reconstitution results that the use of a high ratio of LgnA led to decreased production of **6** and increased hydrolytic product, **12**, probably due to a loss of peptide extension efficiency. A similar biochemical precedent of the stoichiometry of $TE_{II}$ domains and their NRPS partners was also observed in the glycopeptide biosynthesis where the inclusion of a low ratio (2.5 mol%) of $TE_{II}$ domain and its NRPS partner was the most efficient for the editing roles.[31]

Examination of the *lgn* BGC indicated that there are two putative promotor regions, one for *lgn*A and the other for *lgn*B, *lgn*C, and *lgn*D (Supplementary Fig. 27), suggesting that the expression level of LgnA in the producing strain may be different to the ones of LgnB-D. Further in vivo evidence will be required to warrant whether in vitro results are correlated to what the stoichiometry of LgnA with LgnD and LgnB is required for the efficient metabolite production.

**LgnA $TE_{II}$ interacts directly with the LgnD subunit of the NRPS.** Our observations that LgnA shuttles L-Thr residues from LgnB to LgnD is intriguing, and likely requires specific protein-protein interactions (PPIs) at each interface to ensure biosynthetic fidelity. To investigate possible PPIs between LgnA, LgnB and LgnD, native PAGE and ITC analyses were conducted. Native-PAGE analysis of a 1:1 solution of LgnA and *apo*-LgnD resulted in a dominant new band, which migrated further than both LgnA and LgnD alone in the gel (Supplementary Fig. 28a), suggesting that the interaction of LgnA with LgnD caused the change of the overall shape of the complex (and possibly charge exposure). MALDI-TOF MS peptide sequencing analysis confirmed that this band was indeed the protein complex of LgnA and LgnD (Supplementary Fig. 28b).

Titration of LgnA into *apo*-LgnD during isothermal titration calorimetry (ITC) experiments shed light on the thermodynamic traits of their binding (Table 1 and Supplementary Figure 29), with the curve of the heat trace fitting to the one-site binding model. The dissociation constant $K_d$ was calculated to be $2.550 \pm 0.096\,\mu M$, suggesting that LgnA has strong affinity with *apo*-LgnD in the binary complex. To further investigate the exact location of the PPI, a series of titration of LgnA with truncated LgnD was performed. First titration of LgnA with *apo*-LgnD-$C_1$-$T_1$ and *apo*-LgnD-$C_1$-$T_1$-$C_2$ showed that LgnA displays a strong affinity with both truncated proteins with the dissociation constant ($K_d$) values of $1.01 \pm 0.227\,\mu M$ for *apo*-LgnD-$C_1$- $T_1$ (Table 1 and Supplementary Figure 29) and $1.700 \pm 0.223\,\mu M$ for *apo*-LgnD-$C_1$-$T_1$-$C_2$ (Table 1 and Supplementary Figure 29). Next overexpression of truncated LgnD-$C_1$, and LgnD-$T_1$ domains in *E. coli* was carried out (Supplementary Figure 30). Titration of LgnD-$T_1$ domains with LgnA showed strong binding activity with a $K_d$ value of $7.21 \pm 1.61\,\mu M$ (Table 1 and Supplementary Figure 31) while the LgnD-$C_1$ domain displays no binding affinity (Table 1 and Supplementary Figure 31), indicating that LgnA must interact with LgnD-$T_1$ domain to shuttle the L-Thr residue.

Titration of LgnA to LgnB in our ITC analysis showed no binding affinity (Table 1 and Supplementary Fig. 32). However, when the same titration was conducted with microscale thermophoresis (MST) analysis, which uses changes in fluorescence to measure the movement of molecules along local temperature gradients instead of measuring heat release[32], LgnA was found to indeed have a strong affinity toward *apo*-LgnB-$A_1$-$T_0$ didomain with a $K_d$ value of $1.699 \pm 0.813\,\mu M$ (Table 1 and Supplementary Fig. 32), indicating that LgnA also interacts with LgnB for its aminoacyltransfer and hydrolytic activities. Our results suggest that the interaction between LgnA and LgnB is thermodynamically distinct from the other interactions measured by ITC, which might be indicative of higher order conformational changes. No binding affinity was observed when LgnB was titrated with LgnD and truncated LgnD-$C_1$-$T_1$ (Table 1 and Supplementary Fig. 33).

## Discussion

The mechanism of aminoacyl chain translocation elucidated here for the legonindolizidine NRPS has several unique features. First, the activated L-Thr must be transferred by the standalone $TE_{II}$ orthologue, LgnA, from the LgnB-$T_0$ domain to the LgnD-$T_1$ domain for the downstream dehydration reaction. Our observations of a highly specific chain translocation event in NRPSs expands the catalytic scope of the $TE_{II}$ enzyme family beyond canonical (amino)acyl chain hydrolysis. Second, the biosynthetic pathway involves an unusual dehydration reaction on L-Thr that must occur at the $T_1$ domain to provide the Dhb residue. Third, LgnB is an $A_1$-$T_0$ didomain, separated from the core megasynthetase, LgnD. The consequence of this is that the $TE_{II}$ orthologue, LgnA, has been recruited to serve as an adapter to bridge LgnB-$T_0$ and LgnD-$T_1$ domains in order to maintain biosynthetic fidelity.

Standalone $TE_{II}$ domains are a common feature in many of PKS, NRPS and PKS-NRPS hybrid pathways[33]. They play an important role in maintaining the efficiency of the assembly line and enhancing product titres[33]. It has been well-established that $TE_{II}$ domains either remove nonreactive (amino)acyl residues or aberrant intermediates to restore the performance of NRPSs and PKSs, or control the starter units and participate substrate selection[33]. In a few cases, $TE_{II}$ domains release key intermediates[34] or the final products[35]. Our data show that the standalone $TE_{II}$ domain, LgnA, catalyses an unprecedented transaminoacylation reaction that shuttles LgnB-$T_0$-tethered L-Thr to the LgnD-$T_1$ domain of the downstream megasynthetase (Fig. 5a). This is necessary, because the nonselective condensation reaction between IV-CoA species and a L-Thr unit can occur at the LgnB-$T_0$ domain. LgnA hydrolyses the aberrant intermediate, LgnB-$T_0$ tethered IV-Thr to liberate the "blocked" $T_0$ domain (Fig. 5a). Despite the aminoacyltransfer and editing roles played by LgnA, it is possible that the mis-programmed NRPS-tethered IV-Thr (presumably $T_2$-tethered-IV-Thr-Pro), stemming from the nonselective condensation domains, may still accumulate (Fig. 5a), thus blocking the assembly line, as observed in our time-course experiments (Supplementary Fig. 13).

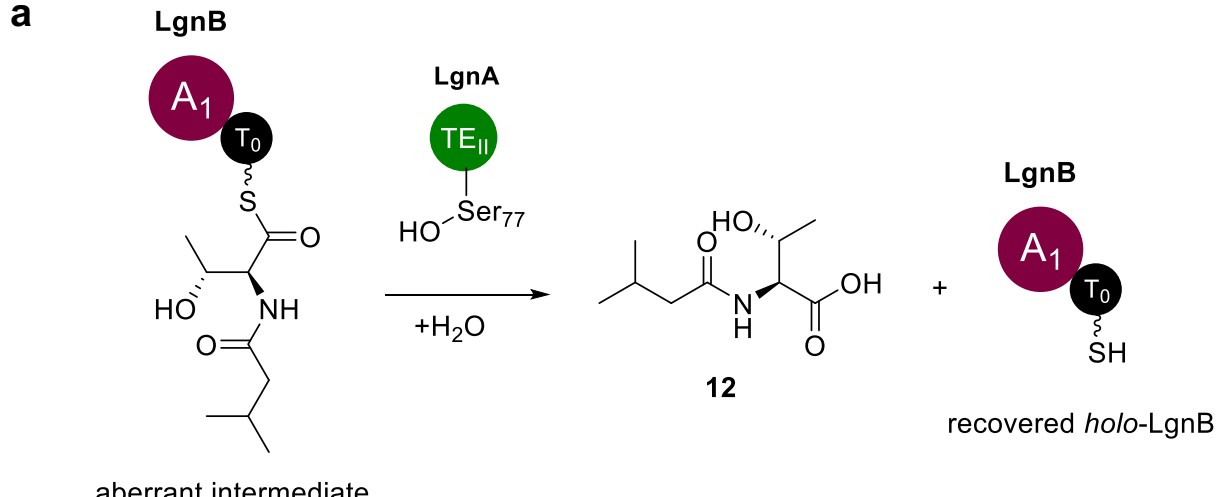

**Fig. 4 LgnA hydrolyses the aberrant intermediate, LgnB-$T_0$-tethered IV-l-Thr. a** A schematic diagram of the hydrolysis of aberrant intermediate catalysed by LgnA. **b** Deconvoluted intact protein mass spectra of (i) *holo*-LgnB $A_1$-$T_0$, (ii) *holo*-LgnD-$C_1$-$T_1$, (iii) an assay of LgnB-$A_1$-*holo*-$T_0$ + *holo*-LgnD-$C_1$-$T_1$ + ATP + L-Thr + $Mg^{2+}$ + IV-CoA, (iv) an assay of LgnB-$A_1$-*holo*-$T_0$ + *holo*-LgnD-$C_1$-$T_1$ + ATP + L-Thr + $Mg^{2+}$ + IV-CoA followed by ultracentrifugation filtration to remove unreacted small molecules and subsequent addition of LgnA-$TE_{II}$, (v) an assay of LgnB-$A_1$-*holo*-$T_0$ + *holo*-LgnD-$C_1$-$T_1$ + ATP + L-Thr + $Mg^{2+}$ + IV-CoA + LgnAS77A-$TE_{II}$ variant. An Peaks labelled with grey dots and asterisks in the intact MS spectra indicate N-terminal acetylation or gluconoylation, respectively, both of which are known post-translational modifications of recombinant heterologous proteins in *E. coli*[24]. The exact measured and observed masses for each species are detailed in Supplementary Table 3.

**Table 1 Summary of ITC and MST assays of domain interactions of LgnA with LgnD, truncated LgnD proteins and LgnB as well as LgnB with LgnD and truncated LgnD proteins.**

| Domain interaction | $K_d$ value | Domain interaction | $K_d$ value |
|---|---|---|---|
| LgnA $\Rightarrow$ LgnD ($C_1$ $C_2$ $A_2$ TE) | 2.55 ± 0.964 µM | LgnA $\Rightarrow$ LgnD-$C_1$ ($C_1$) | 0 |
| LgnA $\Rightarrow$ LgnD-$C_1T_1$ ($C_1$) | 1.01 ± 0.227 µM | LgnA $\Rightarrow$ LgnB ($A_1$) | 1.699 ± 0.813 µM[#] |
| LgnA $\Rightarrow$ LgnD-$C_1T_1C_2$ ($C_1$ $C_2$) | 1.70 ± 0.223 µM | LgnB $\Rightarrow$ LgnD ($C_1$ $C_2$ $A_2$ TE) | 0 |
| LgnA $\Rightarrow$ LgnD-$T_1$ ($T_1$) | 7.21 ± 1.61 µM | LgnB $\Rightarrow$ LgnD-$C_1T_1$ ($C_1$) | 0 |

The $TE_{II}$-mediated transaminoacylation may not be uncommon. For example, WS9326A, a specialised cyclodepsipeptide isolated from various *Streptomyces*, exhibits several interesting chemical features, such as an (*E*)-2,3-dehydrotyrosine and a (*Z*)-pentenylcinnamoyl moiety.[36–38] Analysis of the corresponding *ws* BGC[39] indicated the presence of two standalone upstream modules (WS22 and WS23) consisting of an A-T didomain, two $TE_{II}$ domains (WS5 and WS20, respectively), and the adjacent downstream module 7 with an arrangement of C-T-C-A-T, similar to LgnD. The two A domains in WS22 and WS23 were predicted to activate L-*allo*-Thr and L-Asn, respectively, key chemical moieties of WS9326A[39]. A series of gene mutation experiments have been performed to investigate the biosynthetic logic of this NRPS assembly[39]. Interpretation of the structures of the accumulated products in these variants provided conclusions that two $TE_{II}$ domains, WS5 and WS20, are highly likely to act as shuttling enzymes to translocate the activated L-*allo*-Thr and L-Asn from WS22 and WS23, respectively, to the downstream iterative module 7, which adds these two amino acids to the growing peptide chain. MST analysis revealed considerably high binding affinities between WS5-*apo*-WS23, WS20-*apo*-WS22 and WS20-*apo*-WS23 while relatively low binding affinities between WS5-*apo*-WS22[39]. Crystallographic studies of WS5 and structural modelling suggested possible PPIs between WS5 and the T domains of WS22/23. Our ITC and MST analysis indicated that LgnA displays a strong binding affinity toward the $T_1$ domain of LgnD and LgnB, respectively. More recently, another $TE_{II}$ orthologue, RthD, was proposed to transfer two Thr units into the growing peptidyl chain during the biosynthesis of rotihibins, specialised metabolites with potent herbicidal activity, identified from the phytopathogen *Streptomyces scabies* that is associated with common scab disease[40]. Interestingly, although LgnA displays a very low sequence homologue (<30% AA identity) with WS5, WS20 and RthD, our phylogenetic analysis suggested that WS20 and RthD belong to the same clade of LgnA, while WS5 is grouped into $TE_{II}$ domains that have functions of intermediate releasing or editing (Fig. 5b). However, further biochemical characterization is needed to confirm the exact roles played by these $TE_{II}$ systems in the corresponding NRPS systems.

LgnA was also found to be grouped into a separate clade with other putative LgnA-like $TE_{II}$ domains from the classical standalone $TE_{II}$ enzymes, type I TE domains, aminoacyltransferases, suggesting that LgnA is likely to be diverged from typical $TE_{II}$ domains (Fig. 5b). With this knowledge in hand, we used phylogeny-based, mechanism-guided conserved genomic analysis to identify BGCs containing LgnA-like open reading frames (ORFs), and therefore biosynthetic pathways likely involving this $TE_{II}$-mediated chain transfer. This led to the identification of multiple BGCs encoding multidomain NPRSs and standalone LgnB-like NRPSs (Supplementary Fig. 34). Representatives are distributed in Gram-positive and Gram-negative bacteria, including biologically important plant pathogens: *Pseudomonas fluorescens*, *Xanthomonas arboricola*, nitrogen fixing bacteria such as *Caballeronia choica*, *Hahella* sp. KA22 and *Rheinheimera sp.*; the well-studied model strains, *S. albus* and *S. avermitilis*. Common features in these BGCs are that there are extra C-T didomains located in the middle of one of multidomain NRPSs. In a few cases, the C domains in these C-T didomains belong to the group of C* domains which are predicted to catalyse dehydration reactions (Supplementary Fig. 14). It is likely that LgnA-like aminoacyltransferases transfer amino acid residues activated by LgnB-like NRPSs into the extra T domain of the downstream pathway to incorporate modified amino acid residues (i.e. Dhb units) in the growing peptide chains (Supplementary Fig. 34). Taken together, the recruitment of biosynthetic $TE_{II}$-like transaminoacylation enzymes to biosynthetic clusters may enable the evolution of NRPS pathways by inserting specialised amino acid residues into the growing peptide chains, thereby increasing the diversity of natural product structure.

In the broader context, there have been considerable efforts to use bioengineering approaches to directly modify NRPS machineries in vivo to generate optimized natural products[41]. However, most attempts to achieve this have yielded impaired or non-functional biosynthetic machineries[42]. More recently, a new NRPS domain swapping biotechnology has emerged that uses defined exchange units fused at specific positions that connect two domains and respect the original specificity of the downstream module to enable the production of new peptides[43,44]. In

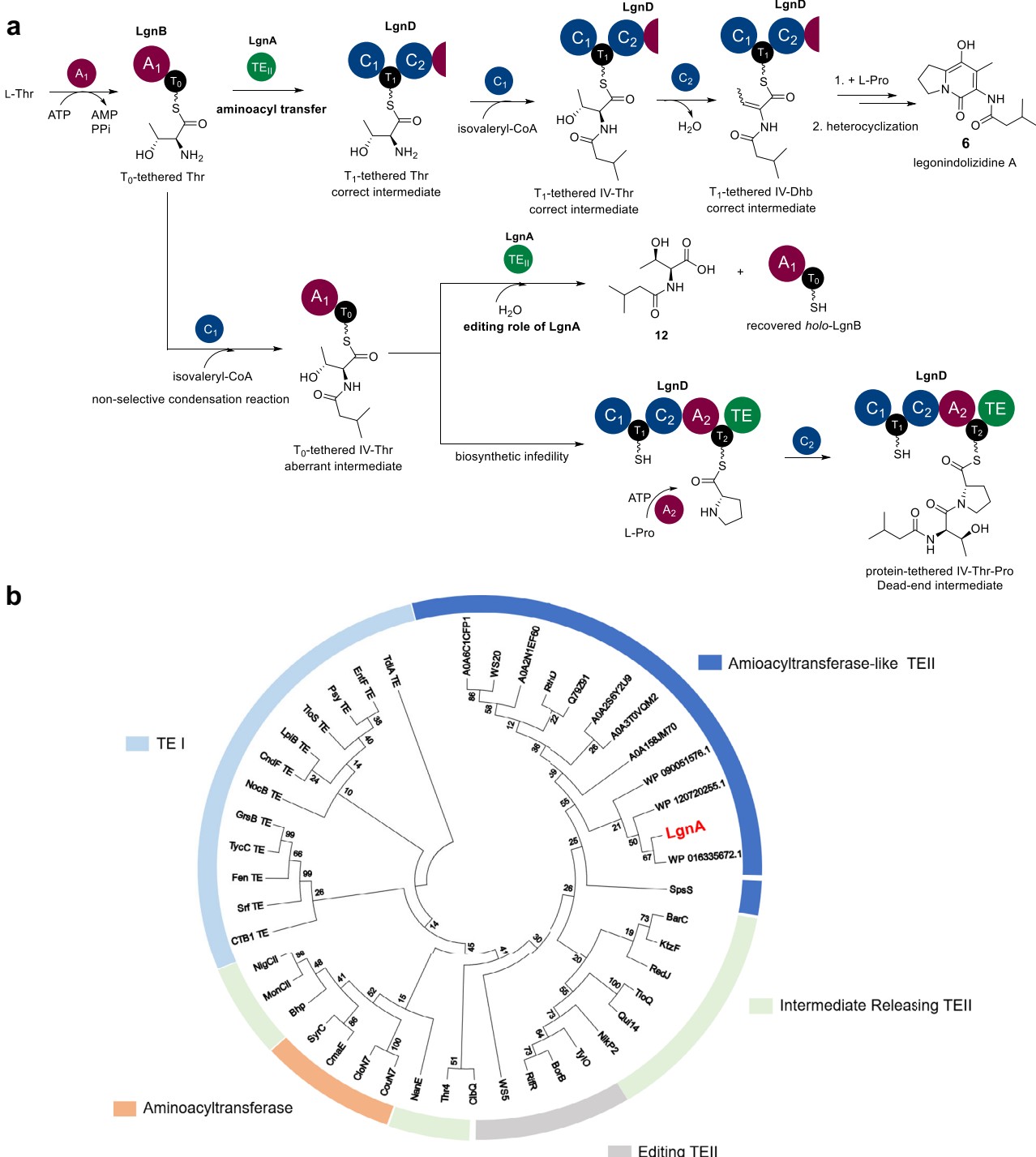

**Fig. 5 Overview of the essential role of the bifunctional TE$_{II}$, LgnA, in the legonindolizidine A 6 NRPS assembly line. a** LgnA catalyses L-Thr unit transaminoacylation reaction between LgnB-T$_0$ and LgnD-T$_1$ domain. The correct intermediate, LgnD-T$_1$-bound L-Thr, undergoes the first condensation with IV-CoA to yield LgnD-T$_1$-bound IV-Thr, followed by the C$_2$-catalysed dehydration on L-Thr residue to provide LgnD-T$_1$-tethered IV-Dhb for the downstream pathway. Due to the nonselective condensation reaction, LgnB-T$_0$-bound L-Thr reacts with IV-CoA to generate the aberrant intermediate, LgnB-T$_0$-bound IV-Thr. LgnA plays the editing role of removing LgnD-T$_0$-bound IV-Thr to regenerate the *holo*-LgnB. However, the nonselective condensation domain could catalyse the reaction between LgnB-T$_0$-tethered IV-Thr and the downstream L-Pro unit to generate the "dead-end" intermediate protein-tethered IV-Thr-Pro. **b** Phylogenetic analysis indicates that LgnA and LgnA-like TE$_{II}$ CDSs form a distinct clade from typical TE$_{II}$ enzymes, TE$_I$ domains, and known aminoacyltransferases in NRPSs. A maximum likelihood tree containing bootstrap values was generated using MEGA 7.0.26. The sequence information for thioesterases and aminoacyltransferases used in this analysis are listed in Supplementary Table 4.

this regard, our finding of LgnA together with its NRPS partner domains may offer a Dhb-forming cassette to generate new Dhb-containing molecules as functionalized peptides, where the Dhb moiety could also provide a bioorthogonal handle[45] for further diversification for peptide-based compound library generation. Further molecular dissection will be required to shed insight on the PPI between LgnA and LgnD, as well as potential exchangeable units present in LgnD-$C_1$-$T_1$-$C_2$ tridomain.

In conclusion, our results provide important insights into a new mechanism of chain translocation from the legonmycin NRPS. We demonstrate that the $TE_{II}$ orthologue, LgnA, is recruited to catalyse an unusual transaminoacylation between T domains for the downstream condensation and dehydration reactions. This is necessitated by the promiscuity of the condensation reactions that can act on the aminoacyl donors when they are attached to the wrong T domain, thus abrogating the assembly line. The occurrence of LgnA-like open reading frames associated with NRPS assembly lines suggest the important biological role to contribute to natural product structural diversity.

## Methods

**General chemicals, reagents, and analytical methods**. All starting materials and reagents were bought from commercial sources and used as received. All biochemical reactions apart from where noted were carried out in triplicate. Before every set of measurements, triplicate control reactions were performed to ensure that the assay were functioning correctly. All flash column chromatography was carried out using silica purchased from Sigma Aldrich using the solvent system noted. $^1H$ NMR spectra were recorded at 400 and 700 MHz using Bruker Avance III and Varian VNMRS-700 spectrometers. $^{13}C$ NMR spectra were recorded at 101, 151, and 176 MHz using Bruker Avance III, Varian VNMRS-600 and Varian VNMRS-700 spectrometers. All coupling constants are reported in Hertz (Hz). Chemical shifts are reported in ppm and are referenced to residual solvent peaks; $CHCl_3$ ($^1H$ 7.26 ppm, $^{13}C$ 77.0 ppm) and $CH_3CN$ ($^1H$ 1.96 ppm, $^{13}C$ 118.3). Mass spectra of synthetic materials were collected on a Waters TQD mass spectrometer and accurate mass spectra were collected on a Waters LCT Premier XE mass spectrometer. Samples were lyophilised using a Christ Alpha LD Plus 1-2.

Enzymatic assays were analyzed on a Bruker MaXis II ESI-Q-TOF-MS connected to a Dionex 3000 RS UHPLC fitted with an ACE C4-300 RP column (100 × 2.1 mm, 5 μm, 30 °C). The column was eluted with a linear gradient of 5–100% MeCN containing 0.1% formic acid over 30 min. The mass spectrometer was operated in positive ion mode with a scan range of 200–3000 m/z. Source conditions were: endplate offset at −500 V; capillary at −4500 V; nebulizer gas ($N_2$) at 1.8 bar; dry gas ($N_2$) at 9.0 L min$^{-1}$; dry temperature at 200 °C. Ion transfer conditions were: ion funnel RF at 400 Vpp; multiple RF at 200 Vpp; quadrupole low mass at 200 $m/z$; collision energy at 8.0 eV; collision RF at 2000 Vpp; transfer time at 110.0 μs; pre-pulse storage time at 10.0 μs.

LC data were analysed using Agilent Chemstation. MS data were analysed using Bruker DataAnalysis or Thermo Xcalibur.

**Synthesis of S-(2-acetamidoethyl) (S)-1-(O-t-butyl-N-(3-methylbutanoyl)-L-threonyl) pyrrolidine-2-carbothioate 10b**. To an SPPS tube was added H-Pro-2-ClTrt resin (0.428 g, 0.30 mmol, 0.7 mmol/g loading) and DMF (5 mL) this was agitated for 1 h to swell the resin. The resin was drained and a solution of Fmoc-Thr(tBu)-OH (0.298 g, 0.75 mmol), PyBOP (0.390 g, 0.75 mmol) and DIPEA (0.194 g, 1.50 mmol) in DMF (5 mL) was added and agitated for 2 h. The resin was drained and washed with DMF (3 × 5 mL). The coupling reaction was then repeated for another 2 h before the resin was once again drained and washed with DMF (3 × 5 mL). To the resin was added a solution of 20% piperidine in DMF (5 mL) and the resin agitated for 30 min. The resin was then drained and washed with DMF (3 × 5 mL) before TEA (0.303 g, 3.00 mmol) in DMF (5 mL) was added followed by isovaleryl chloride (0.362 g, 3.00 mmol) dropwise. The resin was then agitated for 2 h before being drained and washed with DMF (2 × 5 mL) followed by DCM (5 × 10 mL). The resin was then cleaved with a solution of 20% hexafluoroisopropanol in DCM (2 mL) for 1 h. The resin was drained, and the filtrate collected, the resin was washed with DCM (3 × 5 mL) and the filtrates combined. The solution was then concentrated under reduced pressure to give a yellow oil. The residue was taken up in DCM (20 mL) in a round bottomed flask and to the reaction mixture was added N-acetylcysteamine (0.358 g, 3.00 mmol), EDC·HCl (0.115 g, 0.60 mmol), HOBt (0.081 g, 0.6 mmol) followed by DIPEA (0.193 g, 1.50 mmol) and the reaction mixture stirred for 24 h at rt. After this time the reaction mixture was concentrated under reduced pressure and the recovered residue was purified by flash column chromatography (100% DCM to 5% MeOH 95% DCM). This gave the desired compound as a clear oil (0.033 g) in a 4% yield.

$^1H$ NMR (400 MHz, $CDCl_3$) δ 6.61 (d, J = 8.3 Hz, 1H), 6.33 (d, J = 5.4 Hz, 1H), 4.77 (dd, J = 8.4, 3.4 Hz, 1H), 4.72–4.65 (m, 1H), 4.03–3.73 (m, 4H), 3.67–3.44 (m, 4H),

3.36 (ddd, J = 13.9, 8.5, 3.3 Hz, 1H), 3.26–3.14 (m, 1H), 3.07 (t, J = 6.5 Hz, 1H), 2.99–2.94 (m, 1H), 2.87 (ddd, J = 13.9, 6.2, 3.1 Hz, 1H), 2.00 (s, 3H), 1.24 (s, 9H), 1.11 (d, J = 6.4 Hz, 3H), 1.00–0.95 (m, 6H).

$^{13}C$ NMR (101 MHz, $CDCl_3$) δ 201.42, 172.27, 171.17, 168.89, 84.45, 75.05, 69.00, 66.45, 55.54, 46.92, 46.07, 38.73, 31.95, 29.57, 28.36, 27.92, 26.31, 22.78, 22.43, 22.41, 22.11, 18.40.

HRMS ESI$^+$ Calculated for $[M + H]^+$ $C_{22}H_{40}N_3O_5S^+$ = 458.2689 Found = 458.2694

The addition of TFA facilitated the conversion of protected **10b** to IV-Thr-Pro-SNAC **10a** prior to the analysis of thioester exchange in the presence of cysteamine.

**General microbiology and cloning methods**. The primers used in this study are listed in Supplementary Table 1. The genes, *lgnA*, *lgnB*, and *lgnD*, as well as truncated *lgnD* fragments were amplified from cosmid 7G1[19] using KOD hot start DNA polymerase (Novagen). The resulting PCR products were inserted into the *Eco*RI/ *Hin*dIII site of pET-28a(+) in the cases of LgnA, LgnB, and truncated proteins, LgnD-$C_1$-$T_1$ and LgnD-$C_1$-$T_1$-$C_2$ or into the *Nde*I/ *Hin*dIII site of pET-28a(+) in the cases of LgnD-$C_1$ and LgnD-$T_1$ or into the *Nde*I/*Xho*I site of pEHISTEVa in case of LgnD using In-Fusion® HD Cloning Kit (Takara Bio, Inc) or T4 DNA Ligase (Thermo Fisher Scientific). DNA extraction from *E. coli* was carried out using a QIAprep Spin Miniprep Kit (Qiagen). Restriction enzymes (NEB) were used according to the instructions provided by the manufacturers. Site directed mutagenesis of LgnA, LgnD-$C_1$-$T_1$ and LgnD-$C_1$-$T_1$-$C_2$ was achieved using In-Fusion® HD Multiple-Insert Cloning. For LgnA(S77A) variant, two DNA fragments containing the desire mutant site were amplified from the template pET-28a-*lgnA* using a primer pair LgnA_28afor and LgnA_S77toArev, and a primer pair LgnA_S77toAfor and LgnA_28arev, respectively (see Supplementary Table 1 for primer sequences), and cloned into the *Eco*RI/ *Hin*dIII site of pET-28a(+) vector using In-Fusion® HD Cloning Kit. For LgnD-$C_1$-$T_1$(S492A) or LgnD-$C_1$-$T_1$(S492A)-$C_2$ variant, two PCR reactions were performed with a primer pair of LgnD_for and LgnD_T1_StoArev, and a primer pair of LgnD_T1_StoAfor and LgnD_CT1rev or LgnD_C2rev, respectively, using pET-28a-*lgnD*-$C_1T_1$ or pET-28a-*lgnD*-$C_1T_1C_2$ as the template. The resulting products were cloned into pET-28a(+) vector by the method described above. The desired mutation in each variant was verified by DNA sequencing.

The genes were amplified using the PCR primers detailed in Supplementary Table 1.

**General methods of protein expression and purification**. The above corresponding constructs were individually transformed into *E. coli* BL21-CodonPlus (DE3)-RP. Single colonies from each transformation were grown overnight in LB media (5 mL) containing kanamycin (50 μg/mL) and chloramphenicol (25 μg/mL). The overnight culture was transferred to fresh LB medium (500 mL) supplemented with kanamycin (50 μg/mL) and chloramphenicol (25 μg/mL) and cultivated at 37 °C until the cell density reached an $OD_{600}$ of 0.6. IPTG was added to a final concentration of 0.1 mM to induce protein expression. Cells were grown for 16-20 h at 16 °C and then harvested by centrifugation at 4 °C. The cells pellets were resuspended in ice-cold lysis buffer (20 mM Tris-HCl, 300 mM NaCl, 20 mM imidazole, pH 8.0), and further disrupted by Ultrasonic Homogenizer JY92-IIN. Then the supernatant of cell debris was loaded onto Ni-NTA-affinity column. Bound proteins were eluted with the same Tris-HCl buffer containing different concentrations of imidazole. The desired elution fractions were combined and concentrated using a Centrifugal Filter Unit (Millipore). For isothermal titration calorimetry (ITC) measurements of these protein, large scale fermentation (5 L) was conducted and protein solutions from Ni affinity chromatography were further purified by gel filtration chromatography on a Superdex 75 column at 12 °C equilibrated with 20 mM Tris-HCl (PH 8.0) containing 100 mM NaCl. The purified proteins were store at −80 °C in storage buffer (100 mM Tris-HCl, pH 8.0, 150 mM NaCl, 10% (w/v) glycerol, 1 mM DTT). The protein concentrations were determined on NanoDrop 2000 by using the corresponding extinction coefficient.

**In vitro reconstitution of legonindolizidine**. A sample of LgnB (10 μM) and LgnD (10 μM) were incubated with Sfp (2 μM), coenzyme A (100 μM) and $MgCl_2$ (10 mM) in HEPES-Na buffer (50 mM, pH 7.5) at 28 °C for 1 h. To this mixture were added LgnA (10 μM or 0.5 μM), isovaleryl-CoA, L-Thr, L-Pro (1 mM each) and ATP (3.5 mM) to a final volume of 50 μL. The reaction was incubated at 28 °C for 3 h and then quenched by addition of 100 μL of acetonitrile. The mixture was centrifuged at 13,000 rpm for 10 min to remove protein precipitates. The supernatant was then analyzed by Bruker MAXIS II QTOF in tandem with an Agilent 1290 Infinity UHPLC. Samples were separated on a Phenomenex Kinetex XB-C18 (2.6 μM, 100 × 2.1 mm) column with a mobile phase of 5% ACN + 0.1% formic acid to 100% ACN + 0.1% formic acid in 10 min.

**Biochemical determination of NRPS-bound intermediates**. The one-pot in vitro legonindolizidine reconstitution reactions using LgnB (10 μM) and LgnD (or truncated LgnD proteins) (10 μM) with and without the addition of LgnA (0.5 μM) were conducted to capture the biosynthetic intermediates bound to LgnB and LgnD via chemical probe cysteamine. Freshly prepared cysteamine hydrochloride (2.5 μL, 1 M) was added to each reaction (50 μL) as the final component after adding all

other substrates and cofactors. The reaction mixtures were incubated at 28 °C for 3 h and then quenched with 100 μL of acetonitrile. Protein precipitates were removed from the reaction mixtures by centrifugation at 13,000 rpm for 10 min and the supernatants were subjected to LC-MS analysis under the condition as described above.

**Biochemical assay of LgnA hydrolytic activity.** The one-pot in vitro lego-nindolizidine reconstitution reactions using LgnB and LgnD with different concentrations of LgnA (0, 0.31 μM, 0.62 μM, 1.25 μM, 2.5 μM, 5 μM, 10 μM) were conducted to determine the hydrolytic activity of LgnA and the optimal concentration of LgnA used in the one-pot transformation assay. The reaction mixtures were incubated at 28 °C for 3 h, quenched with 100 μL of acetonitrile and were clarified by centrifugation. The supernatants were analysed by Bruker MAXIS II QTOF in tandem with an Agilent 1290 Infinity UHPLC.

Hydrolytic assays of synthetic IV-Thr-SNAC **14a** were initiated by addition of **14a** (1 mM) to a solution of LgnA TE$_{II}$ or LgnA TE$_{II}$ (S77A) (50 μM) in 20 mM Tris, 100 mM NaCl, pH 7.4 (50 μL total volume), and reactions were allowed to proceed for 1 h at 25 °C. The enzyme-free control reaction was achieved by replacing the volume of LgnA TE$_{II}$ with buffer. Each reaction was extracted by addition of 100 μL of EtOAc, vortexed for 30 s, and centrifuged (14,000 g, 2 min). The organic phase was transferred to a glass vial and dried under vacuum. Dried extracts were resuspended in MeOH prior to UHPLC-ESI-Q-TOF-MS analysis.

**LgnB activity assay.** LgnB (10 μM) was converted to its *holo*-form by incubation with Sfp (2 μM), coenzyme A (100 μM) and MgCl$_2$ (10 mM) in HEPES-Na buffer (50 mM, pH 7.5) at 28 °C for 1 h. To this mixture were added L-Thr (1 mM), ATP (3.5 mM) and cysteamine (50 mM) to a final volume of 50 μL. A reaction in the absence of L-Thr was set up as a control. The reaction mixtures were incubated at 28 °C for 3 h. After quenched with of acetonitrile (100 μL), the mixtures were subjected to centrifugation prior to LC-MS analysis.

**LgnD-C$_1$-T$_1$ and LgnD-C$_1$-T$_1$-C$_2$ activity assay.** LgnB (10 μM) and LgnD-C$_1$-T$_1$ or LgnD-C$_1$-T$_1$-C$_2$ (10 μM) were converted to their *holo*-forms as described above. To this mixture were added LgnA (0.5 μM), L-Thr (1 mM), isovaleryl-CoA (1 mM), ATP (3.5 mM) and cysteamine (50 mM) to a final volume of 50 μL. A reaction in the absence of LgnA was set as a control. The reaction mixtures were incubated at 28 °C for 3 h. After quenched with of acetonitrile (100 μL), the mixtures were subjected to centrifugation prior to LC-MS analysis.

**UHPLC-ESI-Q-TOF-MS analysis of intact proteins.** Purified LgnB and truncated LgnD-C$_1$-T$_1$ were converted to their *holo*- forms as described previously. Biochemical measurements were repeated in duplicate for substrates/protein tested. Before every set of measurements, duplicate control reactions were performed to ensure that the assay were functioning correctly.

For the assay of LgnB, loading of L-Thr was initiated by addition of ATP (5 mM), and L-Thr (1 mM) in a final volume of 50 μL. The loading reaction was allowed to proceed (30 min, 25 °C) before intact protein analysis by UHPLC-ESI-Q-TOF-MS.

To detect the condensation activity of LgnD-C$_1$, an assay of LgnB (200 μM) and LgnD-C$_1$-T$_1$ or LgnD-C$_1$-T$_1$(S492A) variant (200 μM) was initiated by addition of ATP (5 mM), L-Thr (1 mM) and isovaleryl CoA (1 mM) in a final volume of 50 μL. The reactions were allowed to proceed (30 min, 25 °C) before intact protein analysis by UHPLC-ESI-Q-TOF-MS.

To detect the transthiolation activity of LgnA or LgnA(S77A) variant, an assay of *holo*-LgnB (100 μM) with addition of LgnA or LgnAS77A variant (200 μM) was initiated by addition of ATP (5 mM) and L-Thr (1 mM) in a 50 μL reaction. The reactions were allowed to proceed (30 min, 25 °C) before intact protein analysis by UHPLC-ESI-Q-TOF-MS.

To detect the hydrolytic activity of LgnA, an assay of LgnB (200 μM) and LgnD-C$_1$-T$_1$ (200 μM) was initiated by addition of ATP (5 mM), isovaleryl CoA (1 mM) and L-Thr (1 mM) for 30 min first in a final volume of 50 μL (30 min, 25 °C). The reaction mixture was transferred to a Centrifugal Filter Unit (Millipore) for ultracentrifugation (13,000 rpm, 5 min). The resultant residue was then topped up with the buffer, followed by addition of LgnA (5 μM) in a final volume of 50 μL for further incubation (30 min, 25 °C) before intact protein analysis by UHPLC-ESI-Q-TOF-MS.

The amino acid sequences of LgnA, LgnB-A$_1$-T$_0$ and LgnD-C$_1$-T$_1$ were listed in Supplementary Table 2. The expected and calculated molecular masses of LgnA, LgnB-A$_1$-T$_0$ and LgnD-C$_1$-T$_1$ were listed in Supplementary Table 3.

**Isothermal Titration Calorimetry (ITC) Measurements.** Thermodynamic data of interaction measurements between LgnA and LgnD or truncated LgnDs were generated using a Malvern MicroCal PEAQ-ITC instrument controlled by MicroCal PEAQ-ITC Control Software. Calorimetric titrations were performed with thirteen 3 μL injections with injection duration of 6.0 s, a spacing of 150 s, a reference power of 5 μcal/s, and constant temperature at 25 °C. The concentrations used for titration experiment were LgnA (125 μM) and LgnD (25 μM) or truncated proteins, LgnD-C$_1$-T$_1$ or LgnD-C$_1$-T$_1$-C$_2$ or LgnD-C$_1$ or LgnD-T$_1$, and LgnB (250 μM) with LgnD or truncated LgnD (25 μM). Every set of measurements was repeated in duplicate. Data were analyzed with the "one set binding model" of the software (MicroCal PEAQ-ITC Analysis) provided by the manufacturer. The software calculates the calorimetric binding enthalpy, adsorption constant, number of adsorption sites, and changes in entropy. Enthalpy of dilution effects for solute was considered during ITC measurements.

**Microscale Thermophoresis (MST) measurement.** To test the binding affinity between LgnA and LgnB, LgnA was utilized as the ligand for the target protein LgnB. The purified LgnB was labelled by RED-tris-NTA and diluted to a final concentration of 25 nM with MST optimized buffer (HBS-ET: 10 mM HEPES, 150 mM NaCl, 3 mM EDTA, 0.005% Tween-20, PH 8.0). LgnA (20 μM) was diluted 1:1 in HBS-ET buffer (10 μL) to generate a 16-ligand serial dilution from 20 to 0.00061 μM. Labelled LgnB (10 μL) was then added to each above LgnA solution (10 μl). After 10 min incubation at room temperature, 16 capillaries were filled with each of samples of which ligand concentrations range from 10 to 0.000305 μM, and were subjected to the MST measurement to yield the respective data points. MST experiments were performed on a Monolith NT.115 instrument (NanoTemper Technologies, Germany) which was set to 80% LED power and 40% MST power at 25 °C. Data analyses were performed using MO. Affinity Analysis v2.3.

**Reporting summary.** Further information on research design is available in the Nature Research Reporting Summary linked to this article.

## Data availability
Data supporting the findings of this work are available within the paper and its Supplementary Information files. A reporting summary for this Article is available as a Supplementary Information file. The datasets generated and analysed during the current study are available from the corresponding author upon request. Amino acid sequence GenBank accession numbers used in this study are reported in Supplementary Table 4 and Supplementary Fig. 14.

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

## Acknowledgements

This work was supported by the Biotechnology and Biological Sciences Research Council UK (S.W. and H.D., BB/P00380X/1 and BB/R00479X/1, W.B. and S.L.C., BB/P003656/1). S.W and H.D. are grateful for SFC Covid-19 Grant extension and bridging Fund and UKRI Covid-19 Extension Allocation Fund. Q.Z. and Y.Y. are grateful for financial support from the National Key Research and Development Program of China (2018YFA0900400), the National Natural Science Foundation of China (31570033, 31811530299, and 31870035 to Y.Y.). H.D. and Y.Y. are recipients of a Royal Society-NSFC Newton Mobility Grant Award (IEC\NSFC\170617). M.J. is the recipient of a BBSRC Discovery Fellowship (BB/R012121/1). The Bruker MaXis II instrument used in this study was funded by the BBSRC (BB/M017982/1). K.K. and H.D. thank Leverhulme Trust-Royal Society Africa award (AA090088) and the jointly funded UK Medical Research Council-UK Department for International Development (MRC/DFID) Concordat agreement African Research Leaders Award (MR/S00520X/1).

## Author contributions

S.W., W.B., S.L.C., M.J., K. K., Y.Y. and H.D. designed the experiments. S.W., Z.L., M-H.T., W.B., Q.Z., K. W. and M.J. performed chemical synthesis, biochemical experiments, small molecule/intact protein mass spectrometry, ITC and bioinformatics analyses. H.D. wrote the paper with S.L.C., K.K., Y.Y., M.J. S.W., W.B. and Q.Z.

## Competing interests

The authors declare no competing interests.

## Additional information

**Peer Review Information** *Nature Communications* thanks Max Cryle and the other, anonymous, reviewer(s) for their contribution to the peer review of this work. Peer reviewer reports are available.

