## [Peer review file · Nature Communications]

Aminoacyl Chain Translocation Catalysed by a Type II Thioesterase Domain in an Unusual Non-Ribosomal Peptide SynthetaseThe authors present the characterization of enzymes involved in the production of legonindolizidine A. They found that LgnA is a TE_{II} protein involved in transferring substrates between NRPS enzymes, releasing stalled intermediates, and modifying a C-domains structure to allow it to perform a dehydration reaction in addition to its normal condensation role. The most important of these results was considered to be that LgnA catalyses the chain translocation between LgnB T₀ and LgnD T₁.

The results of this research will be interesting to researchers in the NRPS field. While the English is mainly clear, it is let down by the paper lacking a logical flow at times and a discussion that seems overly speculative. There are also quite a few minor grammatical errors (lines 46-47 “non-ribosomal peptides synthetase” or line 100 “were predicted to be activate” are typical examples) and the manuscript would benefit from thorough proof-reading.

Major Issue 1: My biggest concern is that the results as a whole do not provide a coherent picture. The authors show LgnA binds tightly to the C-T domains from module 1 of LgnD, and this is required for: (i) dehydration by the second C-domain of LgnD, (ii) removing shunt products during biosynthesis, and (iii) transferring substrates between LgnB and LgnD. Given this, I don't then understand how Supp. Fig. S21 shows the optimal ratio of LgnA to the other enzymes is 1:20 and – more importantly - why increasing the concentration of LgnA to a ratio of 1:1 results in a shunt product. If association between LgnA/LgnD is required for these reactions, then how is this functioning when 19 out of 20 LgnD proteins lack a corresponding LgnA? Even if the answer is that LgnA is mobile and can catalyse multiple partners in real time, why would a 1:1 ratio result in just hydrolysis of the shunt product?

Major Issue 2: The paper jumped around a bit making the core message difficult to follow. Here are some examples.

- Over lines 131 to 193 the authors refer to figure 2a(i), then 2b and 2d. They then go back to explain part (ii) of figure 2a and lastly refer to figure 2c. This back and forth takes a lot of explanation and I feel this section could be at least half the length if combined better. Similarly, lines 200-212 seem like an extension figure 2c (or the same data except extracting a different ion?). It should be combined with figure 2c and the minus proline control placed in the supplementary materials.
- At lines 132-172 the authors show that LgnB and LgnD produce compounds 8a and 9a in the absence of LgnA. Producing these later stage intermediates appears to be evidence against the hypothesis that LgnA is involved in substrate transfer between LgnB and LgnD. However, there is no attempt at an explanation until the section starting at line 270. I think this needs to be addressed at this stage rather than in the final experiments of the paper.
- Lines 284-300: The order I am led through figure 5b is: i, iii, v, ii, iv, vi. Making this a little more confusing is that Supp. Fig. S20 contained opposite lettering and appears to have essentially the same content.

Minor issues that need to be addressed:

Line 52: If the authors are going to adopt the nomenclature T domain in preference to PCP domain then it is more important that they define this abbreviation (T = “thiolation”) than the PCP abbreviation they provide.

Lines 193-196: The tree needs to be bootstrapped and the building method should be described in the legend of the supplementary figure.

Lines 233-239 and Supp. Fig. 15: This result does not provide sufficient evidence to conclude LgnA is removing aberrant intermediates. There is no mention of replication making it impossible to determine if the difference is due to experimental variation or a true effect. Nonetheless, the result would be expected without a thioesterase activity. The cystamine in this assay is known to remove intermediates from the NRPS and a decrease of 9a would be expected due to dehydration of Thr in the presence of LgnA (as per figure 2c). At minimum, this possibility should be acknowledged and the number of replicates presented.

Lines 239 – 242: These lines state that LgnA removes the shunt product IV-Thr from the assembly line to stop its accumulation. However, IV-Thr has accumulated in Fig 5b (vi) in the presence of LgnA. How do the authors explain its accumulation given that they earlier showed LgnA removes IV-Thr?

Lines 246-269: The purpose of this paragraph is suggested to be determining whether association of LgnA with LgnD causes a conformational change of LgnD-C₂ that results in the hydration function becoming activated. I don't think the evidence is strong enough to answer this question. The LgnD-C₂ domain was already shown to be catalytically active for condensation, which raises the question of which conformational changes retain that activity but are essential for hydration? I also found it strange that LgnA was shown to bind a different module of LgnD, and as noted above, that it was only needed in a 1:20 ratio.

Fig. 5 and Supp. Fig. 20: I think the “-“ LgnA diagram should be removed or added to the remaining parts of the diagram. It is not shown in parts i-iii of Fig. 5 and the inclusion of the LgnA schematic here gives the impression that the sample contained LgnA – a reader really has to look hard to see the “-“, and even then the difference with i-iii is distracting.

Lines 301 – 306: The authors state Fig. 5b (vi) shows that the condensation between L-Thr and IV-CoA catalysed by LgnD C₁ occurs on LgnD-T₁. However, the same figure shows IV-Thr-CoA attached to LgnB-T₀. They suggest it might be an artefact based on the experiment conditions at lines 310-312. However, couldn't this artefact be the other way around with the opposite happening, i.e. that IV-Thr-CoA is forming on LgnB-T₀ and then being transferred to LgnD-T₁?

Lines 403-406: The phylogenetic tree is unrooted, not many of the branches are supported by the bootstraps and the tree-building method is not stated. These issues need to be addressed and unless the authors create a rooted tree, they should not infer where the common ancestor would be.

Lines 407-430: This paragraph is a bit speculative and does not provide support to the paper.

Lines 450-461 and lines 37-40: It is suggested that LgnA and its partner domains offer a flexible Dhb-forming cassette. There is no evidence to suggest this could be used as a cassette with any other enzyme and seems overly speculative given the competing reactions of LgnA.

Sequences: The authors appear to not have provided sequences. I could not find an accession number for LgnA, LgnB or LgnD.

Reviewer #2 (Remarks to the Author):

The paper by Wang et al describes a new function for a type II TE acting as aminoacyl transferase between different NRPS parts. The paper is really well written, contains a lot of data and supplementary information to back up all findings.

However, I have a few suggestions to improve it further:

- Intact protein MS – is there binding of IV-Thr/Thr to TEII? Maybe via an active site cysteine? Does this intact protein mass measurement allow observation of the assumed interaction of LgnA with LgnD?
- Line 93-95: mention of methyl group in legonmycins. Is this relevant? Then please explain why. If the intention is to point out that they are different from the other Pas maybe point out, that they legonmycins feature additional methyl group due to the incorporation of thr instead of ser?
- Supplementary Figure 15: internal standard for quantification to be able to tell about “trace amounts”? The increase over time for 9a seems to be proportional for both assays, with or without LgnA only the “overall” amount appears to differ. (Supplementary, Figure 15 Line 173)
- Line 268 ITC with LgnA titration to only LgnD C1 or only LgnD T1 could allow even more precise understanding of the interaction
- personally, the sparsomycin section feels a little bit long with respect to the overall low relevance and sequence identity of LgnA with SpsS (Line 407..)
- can the authors comment on the differences of these C* domains for dehydration and which aa might be involved in this catalysis?
- line 321: what could be a mechanisms to ensure this ration in vivo? Can the authors comment on the BGC and possible promoters/regulatory mechanisms?
- typos

Line 60 through or via or both?

Line 75 word missing? ATs catalyse transfer/condensation of

Line 100 to be active vs to be activating vs to activate

Line 101 legonmycin A (5)

Line 217 is LgnA but should be LgnB

Line 402 aminoacyltransferases that mediates

Line 422 Introduction

Line 425 facilitateS

Line 440 common features is

Line 467 when they attached (leave they or add are)

Line 470 contribute to

Line 783, Figure 2: compound 5 (depicted in chromatogram 2a) but not structure for 5 (legonmycin) in figure.

Line 783, Figure 2: indicate cysteamine offloading for the arrows (clearer distinction to the arrows for proposed biosynthetic pathway).

Line 817 Figure 6 n missing in aminoacyltransferase-like TEII

Supplementary

Line 173 Figure 15

Line 193 protein complexes contain

Line 231 Figure a: x-axis incomplete and stacking of graphs?

Line 232 figure caption: a and b mixed. In figure NRPS final product legonindolizidine but nomenclature for 5 (legonmycin)

Line 260 Figure S24 please specify what ? domains are (predicted but unknown domains?)

Line 261 Figure S24 maybe add LgnA-D domain and cluster organisation to the figure? Producing strain and architecture of LgnA-D (including) LgnC are to my knowledge otherwise not mentioned in this article

Reviewer #3 (Remarks to the Author):

This is a very interesting manuscript but also one that is very confusing for the reader to follow and to understand. I really struggled to follow all the experiments and the logic flow within the manuscript. There were many mistakes (particularly in the SI but also in the main text and

figures) that made following the experiments very challenging, and coupled with the lack of information in the captions, missing graph axes etc. I have to admit that I got lost on many occasions even after multiple readings. Given this, I relied heavily on the figures to try and follow the manuscript, however even here I was lost on many occasions - apologies if my comments below therefore miss an experiment that is already included. Citation of the recent ACIE paper showing type II TE transfer should be included in a revised version.

I do feel that the manuscript often relies on assertion and does not always seek to explore other possible answers for the results that are seen (e.g. results in SI Figs 15 and 16).

Starting with figure 2, the caption does not describe what is actually in the reaction, which makes it hard to follow from the beginning. Where is intermediate 8 bound, is it T0 or T1? What does the control reaction look like without proline?

What are the levels of the intermediates shown in panels B and C? These look very similar - shouldn't there be an accumulation of 9 (9a) in the sample without LgnA? I see this in SI Figure 15, but these traces do not have any indication of a Y-axis at all and these traces do not seem to correspond with the traces in Figure 2. The experiment is also somewhat flawed, as the role of LgnA in hydrolysis of the "incorrect" intermediate cannot be made as this enzyme is also apparently required for its formation - this could also be the role of the main TE domain. Also for SI Figure 16, I do not see how this statement can be made, given that these intermediates will apparently be loaded onto the TE domain - this could just be hydrolysis over time due to multiple cycles of catalysis leading to higher signal with the same percentage of loss per cycle.

Figure 5 is missing further controls - add IV-CoA and use C1 in its apo form to see if IV is added; then also separate the C1T1 construct with Thr (or load it) to test that.

A key experiment - showing the LgnA loaded with an intermediate - is curiously absent. It would be very nice to be able to show this and to show which species it is (ie just Thr or also IV-Thr).

I think the system needs more controls to really be able to tease out the pathway. In particular, this means S2A mutants for T1 (LgnD_S2A_T1) and T2 (LgnD_S2A_T2) within the LanD construct to be able to tell where different intermediates are during the reaction. This would allow the steps to be broken down and then introduced to the reader in a step-wise manner, rather than in the rather overwhelming way it is currently done. Use of these two LanD constructs in the experiments shown in figure 2 would then allow the early steps to be characterized. This would also be important in Figure 3 where the site of dehydration is not known. It also links to the experiments in figure 5, where the site of addition of the acyl group is unclear (it would be on T0 or T1). Use of a synthetic probe on the T2 Ala mutant would allow the formation of alkene to be probed.

Mutation of the C1 and C2 active sites are also important controls to be able to show the involvement of each of these domains in the biosynthesis process, which is key given the important probably role of the type II TE in aminoacyl transfer.

Figure 4 - the attempt to quantify affinities is commendable, but I feel this should be extended to the LgnD-LgnB and LgnA-LgnB systems as well to understand these interactions as well - the expts in SI Fig 18 appear to show not interaction, but this is likely due to their lack of loading - otherwise how can this TE domain transfer the units between PCP domains? The investigation of the truncated LgnD constructs is interesting but confusing - what is the interaction of the isolated T1 domain with LgnA? One could imagine the other results occur due to sterics and conformations in solution, but knowing the baseline T1 interaction would be very helpful. And are these holo or apo constructs? Again, lack of labeling in the captions make this harder to follow.

Carefully check figures - there are a number with spelling mistakes in them. So many of the labels in the SI figures appear to be wrong - it is really hard to tell what constructs are being used with the protein and domain labels being switched around!

Figure SI 21 is very important - did the assays in the paper all use these optimal conditions (i.e

low LgnA) ratio? These type of experiments have precedent in the last few years for other NRPS systems, although I did not see these cited here.

Authors' responses to the reviewers:

We would like to thank the reviewers for their suggestions and comments. Below we provide an overview of the additional experimental evidence (E) (highlighted in blue) to support our conclusion that LgnA is a new aminoacyltransferase-like type II thioesterase (TE_{II}). We have also included a point by point response to all of the comments raised by each reviewer.

E1. We generated two variants, LgnD-C₁-T₁(S492A) and LgnD-C₁-T₁(S492A)-C₂, where the key Ser residue (Ser492) of LgnD-T₁ was changed to Ala. This change precludes either of these mutants becoming the *holo* form. Incubation of *apo*-LgnB-A₁-T₀ with Thr, Sfp, CoA, IV-CoA, ATP in the presence of the truncated LgnD-C₁-T₁(S492A) mutant resulted in the accumulation of only the IV-Thr cystamine adduct as observed in the chemical capturing analysis (Figure 2d and Supplementary Fig 16). This result indicates that IV-Thr must be loaded in T₀ domain and C₁ is a promiscuous C domain which can recognise T₀-tethered Thr for the condensation reaction with IV-CoA. Compared to the cystamine adducts generated from the assays of LgnB-A₁-T₀ and LgnD-C₁-T₁-C₂ or LgnD-C₁-T₁(S492A)-C₂ with substrates in the presence or absence of LgnA, we concluded that C₂ domain catalyses the dehydration reaction on LgnD-T₁-tethered IV-Thr to generate LgnD-T₁-tethered IV-Dhb unit for the downstream pathway (Figure 2d and Supplementary Fig 17).

E2. Intact MS analysis demonstrated that LgnA was indeed acylated with a Thr residue. When *apo*-A₁-T₀-LgnB was incubated with Thr, Sfp, CoA and ATP in the presence of LgnA acylation was observed. This indicates that LgnA functions as an aminoacyl transferase by capturing the Thr residue from LgnB-A₁-T₀-tethered Thr as a transit intermediate (Figure 3). Mutating Ser₇₇ in LgnA to Ala provided the LgnAS77A variant which completely abolished the final product, legonindolizidine, as well as the key intermediates, IV-dehydrobutyryne (Dhb) and IV-Dhb-Pro units, indicating that Ser77 is the key residue for product formation.

E3. We have generated LgnB-A₁-T₀ tethered isovaleryl (IV)-Thr *in situ* by incubation of *apo*-A₁-T₀-LgnB with Thr, Sfp, CoA, IV-CoA, ATP in the presence of the truncated LgnD-C₁-T₁ didomain, followed by filtration of small molecules such as CoA, IV-CoA and ATP by ultracentrifugation. We clearly observed a fully acylated T₀ domain with and appended IV-Thr unit. Addition of LgnA resulted in the appearance of *holo*-LgnB, strongly suggesting that LgnA hydrolyses IV-Thr unit (Figure 3). We didn't observe any trace of LgnD-C₁-T₁ tethered IV-Thr, only *holo*-LgnB-A₁-T₀ and LgnD-C₁-T₁ were detected. Taken together, our intact MS analysis and site directed mutagenesis coupled with biochemical reactions strongly support that LgnA plays an editing role of removing the aberrant intermediate (T₀-tethered IV-Thr unit) generated from a non-selective condensation reaction.

While the work in E1-3 was being carried out, an attempt of chemoenzymatic synthesis of IV-Thr-pantetheine arm was undertaken. Unfortunately, the attempted coupling of IV-Thr-COOH and pantetheine was unsuccessful (please see the detailed responses at the end of this document). Instead, we synthesised an IV-Thr-SNAC derivative (Supplementary Fig. 23). The assays of IV-Thr-SNAC with LgnA or the LgnAS77A variant, however, indicated that the synthetic compound remained intact during the course of the reactions.

E4. Examination of the *lgn* BGC indicated that there are two putative promoter regions, one for *lgnA* and the other for *lgnB*, *lgnC* and *lgnD* (Supplementary Fig. 27), suggesting that the expression level of LgnA in the producing strain may be different to the ones of LgnB-D. Further *in vivo* evidence will be required to warrant whether *in vitro* results are correlated to what the stoichiometry of LgnA with LgnD and LgnB is required for the efficient metabolite production.

E5. We have overexpressed new truncated recombinant proteins, C_1 and $apo-T_1$. ITC analysis demonstrated that $apo-T_1$ displays excellent binding affinity toward LgnA (7.0 μ M) (Figure 4) while C_1 has no binding affinity toward LgnA (Supplementary Fig. 29). Given that LgnA showed no binding affinity toward LgnB in our ITC analysis, we turned our attention to microscale thermophoresis analysis (MST) which can be used to analyse weak conformation changes. Indeed, our MST assay indicated that LgnA shows affinity toward LgnB, a similar observation to which was reported in the recent paper in *Angew Chemie Int. Ed* (2021, 10.1002/ange.202103872) where the authors reported that two TE_{II} domains, WS5 and WS20, displayed strong affinity toward the upstream NRPS didomains, WS22 and WS23, in the biosynthesis of the cyclodepsipeptide WS9326A. It is worth noting that these two TE_{II} domains were deduced to play a similar role of aminoacyl transfer of L-*allo*-Thr and L-Asn, respectively, to the one of LgnA, based on the interpretation of the structures of the metabolites isolated from genetic variants. Further biochemical investigation is needed to confirm how these two TE_{II} domains transfer the corresponding amino acid unit to the corresponding downstream NRPSs.

Please see below, in blue, our detailed responses to the individual reviewers' comments.

Reviewer #1:

The authors present the characterization of enzymes involved in the production of legonindolizidine A. They found that LgnA is a TE_{II} protein involved in transferring substrates between NRPS enzymes, releasing stalled intermediates, and modifying a C-domains structure to allow it to perform a dehydration reaction in addition to its normal condensation role. The most important of these results was considered to be that LgnA catalyses the chain translocation between LgnB T₀ and LgnD T₁. The results of this research will be interesting to researchers in the NRPS field. While the English is mainly clear, it is let down by the paper lacking a logical flow at times and a discussion that seems overly speculative. There are also quite a few minor grammatical errors (lines 46-47 "non-ribosomal peptides synthetase" or line 100 "were predicted to be activate" are typical examples) and the manuscript would benefit from thorough proof-reading.

We apologize that the structure of the discussion in the manuscript may have led to a misunderstanding about the focus and key findings of our work. The main focus of this paper was to report on a new function of LgnA (TE_{II}) enzyme as an aminoacyltransferase. LgnA transfers Thr from the LgnB-T₀ domain to the T₁ domain, followed by the condensation reaction with isovaleryl (IV)-CoA to generate a LgnD-T₁ tethered-IV-Thr intermediate for the downstream dehydration reaction catalysed by the C₂ domain. The observation from our native Page suggested that LgnA strongly interacts with LgnD and as such the protein complex moves faster than both individual LgnA or LgnD. We have tried our best to change the relevant context in the revised MS to make the focus of this paper clearer. It is not our intention to suggest that the binding of LgnA led to the modification of the C domain to being able to conduct the dehydration. Further structural biology and chemical biology tools such as carbene footprinting technology would allow us to determine the exact nature of the C domains behaviour in the presence or absence of LgnA. These areas will be the focus of our future research efforts, but we feel that they lie outside the scope of this present manuscript.

We have also addressed the relevant grammatical errors highlighted by the reviewer.

Major Issue 1: My biggest concern is that the results as a whole do not provide a coherent picture. The authors show LgnA binds tightly to the C-T domains from module 1 of LgnD, and this is required for: (i) dehydration by the second C-domain of LgnD, (ii) removing shunt products during biosynthesis, and (iii) transferring substrates between LgnB and LgnD.

In order to clarify the overall narrative in the manuscript we have reorganised the original material and also supplemented this with new experimental data (please see summary above). In restructuring the manuscript we believe that it is now clear that LgnA catalyses the aminoacyl transfer of the Thr unit from the T₀ domain to the T₁ domain so that LgnD-T₁-tethered Thr can undergo the condensation reaction with IV-CoA to generate the correct intermediate (LgnD-T₁-tethered IV-Thr) for the downstream pathway. In the meantime, LgnA plays an editing role of removing the aberrant intermediate (LgnB-T₀-tethered IV-Thr) generated from the non-selective condensation reaction. However, due to biosynthetic infidelity, some of the LgnB-T₀-tethered IV-Thr can escape the hydrolysis and as a result the second non-selective condensation could occur to generate the dead-end intermediate (NRPS-tethered IV-Thr-Pro) as observed in our time-course experiments.

Given this, I don't then understand how Supp. Fig. S21 shows the optimal ratio of LgnA to the other enzymes is 1:20 and –more importantly - why increasing the concentration of LgnA to a ratio of 1:1 results in a shunt product. If association between LgnA/LgnD is required for these reactions, then how is this functioning when 19 out of 20 LgnD proteins lack a corresponding LgnA? Even if the answer is that LgnA is mobile and can catalyse multiple partners in real time, why would a 1:1 ratio result in just hydrolysis of the shunt product?

Based on our experimental evidence, it was concluded that the high molecular ratio of LgnA results in the hydrolysis of long-lived LgnD-T₁-tethered IV-Thr, thus reducing the peptide extension efficiency. A similar biochemical precedent was also observed in the literature in glycopeptide biosynthesis where the inclusion of a low ratio of a TE_{II} domain and its NRPS partner was the most efficient for the editing role (*Chem Sci*, 2019, 10, 9466-9482).

Over lines 131 to 193 the authors refer to figure 2a(i), then 2b and 2d. They then go back to explain part (ii) of figure 2a and lastly refer to figure 2c. This back and forth takes a lot of explanation and I feel this section could be at least half the length if combined better. Similarly, lines 200-212 seem like an extension figure 2c (or the same data except extracting a different ion?). It should be combined with figure 2c and the minus proline control placed in the supplementary materials.

As suggested, we have combined figures 2 and 3 to provide the new figure 2 corresponding to our new experiments and re-organised intact MS figures. For consistency and reader fluency, we have also changed the numbers of the chemical adducts as shown Figure 2b. We have also changed the numbers of the traces according to the text flow. We have also moved some figures to the SI document in accordance with the reviewers' suggestions.

At lines 132-172 the authors show that LgnB and LgnD produce compounds 8a and 9a in the absence of LgnA. Producing these later stage intermediates appears to be evidence against the hypothesis that LgnA is involved in substrate transfer between LgnB and LgnD. However, there is no attempt at an explanation until the section starting at line 270. I think this needs to be addressed at this stage rather than in the final experiments of the paper.

We believe that the production of the IV-Thr-Pro intermediate is related to biosynthetic infidelity. As suggested by the reviewer we have added a discussion on this point directly after we report this observation in the revised manuscript. Although LgnA can efficiently transfer Thr from the LgnB-T₀ to the LgnD-T₁ domain for the downstream process (i.e. dehydration), the aberrant intermediate (LgnB-T₀-tethered IV-Thr) could escape the hydrolytic activity of LgnA due to biosynthetic infidelity. This aberrant intermediate can be condensed with the downstream Pro residue to yield NRPS-tethered IV-Thr-Pro.

Lines 284-300: The order I am led through figure 5b is: i, iii, v, ii, iv, vi. Making this a little more confusing is that Supp. Fig. S20 contained opposite lettering and appears to have essentially the same content.

We have re-arranged the order of the figures and addressed the numbering within them as requested.

Line 52: If the authors are going to adopt the nomenclature T domain in preference to T domain then it is more important that they define this abbreviation (T = "thiolation") than the T abbreviation they provide.

As requested, we have changed all PCP or protein carrier protein to thiolation (T) for consistency.

Lines 193-196: The tree needs to be bootstrapped and the building method should be described in the legend of the supplementary figure.

The method for phylogenetic analysis was detailed in the legend. The tree was an output from NaPDoS, which is a commonly used website for the prediction of C domain subtypes. It does not include bootstrap values but have confidence values including in the Newick format output. As requested, we visualized these values using Interactive Tree Of Life (iTOL) (Supplementary Fig. 14).

Lines 233-239 and Supp. Fig. 15: This result does not provide sufficient evidence to conclude LgnA is removing aberrant intermediates.

The observation of hydrolytic product IV-Thr-COOH in the assay with LgnA shows that LgnA has hydrolytic activity towards T-tethered-IV-Thr. To provide further evidence that LgnA is removing aberrant intermediates we have carried out additional experiments and the data arising from these have been added to the revised manuscript. Based on our new evidence of intact MS and site-directed mutagenesis (E3), we hoped that we have provided sufficient evidence that LgnA plays an editing role of removing the aberrant intermediate.

There is no mention of replication making it impossible to determine if the difference is due to experimental variation or a true effect.

We apologise for not including more detail on this point. All of the experiments presented in this paper were conducted in triplicate and this statement is now included in the Materials and methods section.

Nonetheless, the result would be expected without a thioesterase activity. The cystamine in this assay is known to remove intermediates from the NRPS and a decrease of 9a would be expected due to dehydration of Thr in the presence of LgnA (as per figure 2c). At minimum, this possibility should be acknowledged and the number of replicates presented.

During the chemical capturing experiments we conducted, the same conditions in the assays with or without the addition of LgnA were used. in triplicate to avoid potential artefacts (i.e. potential hydrolysis of NRPS-bound intermediates). In our time course experiments, the ion intensities of IV-Thr-Pro was significantly larger in the absence of LgnA than in the presence of LgnA.

Lines 239 – 242: These lines state that LgnA removes the shunt product IV-Thr from the assembly line to stop its accumulation. However, IV-Thr has accumulated in Fig 5b (vi) in the

presence of LgnA. How do the authors explain its accumulation given that they earlier showed LgnA removes IV-Thr?

Based on our previous evidence plus the new experiments conducted, it is believed that LgnB-T₀-tethered IV-Thr is the aberrant intermediate and LgnD-T₁-tethered IV-Thr is the correct intermediate. It is worth noting that the majority of LgnD domains are absent in our intact MS analysis, causing LgnD-T₁-tethered IV-Thr to be stalled. The molecular weight of intact LgnD is beyond the intact MS limit of detection. When increasing the concentration of LgnA, the stalled LgnD-T₁-tethered IV-Thr will also be hydrolysed.

Lines 246-269: The purpose of this paragraph is suggested to be determining whether association of LgnA with LgnD causes a conformational change of LgnD-C2 that results in the hydration function becoming activated. I don't think the evidence is strong enough to answer this question. The LgnD-C2 domain was already shown to be catalytically active for condensation, which raises the question of which conformational changes retain that activity but are essential for hydration?

The main objective is to report the biochemical characterization of LgnA as the first biochemically characterized example of TE_{II} orthologues acting as an aminoacyltransferase-like enzyme. We apologise for any misunderstanding caused and it is not our intention to suggest that the binding of LgnA led to modification of the C domain to display dehydration activity. Further structural biology and chemical probing such as carbene footprinting technology would allow us to determine the exact nature of C domain behaviour in the presence or absence of LgnA. As previously stated we feel that this work lies outside the scope of the current study. We have also changed the relevant context accordingly.

Fig. 5 and Supp. Fig. 20: I think the “-“ LgnA diagram should be removed or added to the remaining parts of the diagram. It is not shown in parts i-iii of Fig. 5 and the inclusion of the LgnA schematic here gives the impression that the sample contained LgnA – a reader really has to look hard to see the “-“, and even then the difference with i-iii is distracting.

We have changed the diagram as requested.

Lines 301 – 306: The authors state Fig. 5b (vi) shows that the condensation between L-Thr and IVCoA catalysed by LgnD C1 occurs on LgnD-T1. However, the same figure shows IV-Thr-CoA attached to LgnB-T0. They suggest it might be an artefact based on the experiment conditions at lines 310- 312. However, couldn't this artefact be the other way around with the opposite happening, i.e. that IV-Thr-CoA is forming on LgnB-T0 and then being transferred to LgnD-T1?

Given the new experiments, it is concluded that the correct assembly should be:

1. LgnB activates Thr and loads it in LgnB-T₀;
2. LgnA transfers Thr from LgnB-T₀ to LgnD-T₁;
3. C₁ of LgnD catalyses the first condensation reaction in LgnD-T₁ to yield LgnD-T₁-tethered-IV-Thr;
4. C₂ of LgnD then catalyses the dehydration reaction in LgnD-T₁ to yield LgnD-T₁-tethered-IV-Dhb;
5. The dehydration on IV-Thr must occur on the LgnD-T₁ domain.

LgnA will hydrolyse the aberrant intermediate LgnB-T₀-tethered IV-Thr as the promiscuous condensation reaction can also act on LgnB-T₀-tethered Thr to yield LgnB-T₀-tethered IV-Thr. Due to biosynthetic infidelity, some of the aberrant intermediate can escape the LgnA-based

hydrolysis and can be condensed with the downstream Pro to generate the dead-end intermediate (NRPS-tethered IV-Thr-Pro).

Lines 403-406: The phylogenetic tree is unrooted, not many of the branches are supported by the bootstraps and the tree-building method is not stated. These issues need to be addressed and unless the authors create a rooted tree, they should not infer where the common ancestor would be.

We agree that the interpretation of the phylogenetic analysis is too over-interpreted. The purpose of the tree analysis was to show that LgnA is grouped into a specific cluster of putative TE_{II}s which are likely to have the similar functions. We have therefore changed the relevant context.

Lines 407-430: This paragraph is a bit speculative and does not provide support to the paper.

We have changed the relevant discussion context of two putative aminoacyltransferase-like TE_{II}s reported in the recent ACIE paper (ACIE, 2021, doi.org/10.1002/anie.202103872) published online during the reviewing stage of our manuscript. This paper described genetic manipulation of a NRPS pathway. The interpretation of the structures isolated from the mutants provided a conclusion that two putative TE_{II} enzymes, WS5 and WS20, are likely to act as aminoacyltransferases to shuttle protein-tethered L-Asn and L-*allo*-Thr residues, respectively. MST analysis indicated that the upstream T domain in A-T didomain NRPS has strong binding affinity with the downstream TE_{II}. However, further biochemical characterization would be needed to consolidate this conclusion.

Lines 450-461 and lines 37-40: It is suggested that LgnA and its partner domains offer a flexible Dhb forming cassette. There is no evidence to suggest this could be used as a cassette with any other enzyme and seems overly speculative given the competing reactions of LgnA.

We accept the reviewers' comments on this point and we have edited the text in the manuscript accordingly.

The authors appear to not have provided sequences. I could not find an accession number for LgnA, LgnB or LgnD.

We had already deposited the gene cluster in our previous ACIE paper (2015, 54, 12697-12701). The accession numbers of LgnA, B and D are AIZ66876.1, AIZ66877.1 and AIZ66879.1, respectively. We have also added these accession numbers in table S2.

Reviewer #2:

Intact protein MS – is there binding of IV-Thr/Thr to TEII? Maybe via an active site cysteine? Does this intact protein mass measurement allow observation of the assumed interaction of LgnA with LgnD?

In our new experimental evidence, we indeed observed an LgnA-bound Thr intermediate (please see E1). We also generated one LgnA variant where the Ser₇₇ residue in LgnA was mutated into Ala. Mutation from S to A in LgnA-S77A completely abolished its activity. Our ITC analysis indicated that LgnA binds with LgnD-T₁ domain to deliver the Thr unit. Unfortunately, the molecular weight of LgnD is too large and beyond the limit of detection of intact MS. We didn't observe the interaction of LgnA and the truncated didomain, LgnD-C₁-T₁, presumably due to the harsh ionisation environment in intact MS.

Line93-95: mention of methyl group in legonmycins. Is this relevant? Then please explain why. If the intention is to point out that they are different from the other Pas maybe point out, that they legonmycins feature additional methyl group due to the incorporation of thr instead of ser?

The reviewer raises an interesting point. We think that the extra methyl group in legonmycin is the key difference to other PAs as the formed key Dhb-containing intermediate is different to the ones in other PA systems. We have modified the relevant text in the introduction.

internal standard for quantification to be able to tell about “trace amounts”? The increase over time for 9a seems to be proportional for both assays, with or without LgnA only the “overall” amount appears to differ. (Supplementary, Figure 15 Line 173)

We apologize that the description in the manuscript may have led to confusion on this point. The purpose of this comparison was to show that the presence of LgnA in the assay indeed dramatically changes the accumulation of the shunt intermediate compared to what happened in the absence of LgnA. We have added the y axis of ion intensities of these LC-MS traces in order to compare the accumulated shunt products between both reaction conditions. We have also changed the relevant context in order to accurately reflect what our intention is.

ITC with LgnA titration to only LgnD C1 or only LgnD T1 could allow even more precise understanding of the interaction.

As suggested we have carried out additional experiments (e.g. E5) to address this point. In summary LgnA has a strong interaction with LgnD-T₁ but not with LgnD-C₁.

personally, the sparsomycin section feels a little bit long with respect to the overall low relevance and sequence identity of LgnA with SpsS (Line 407..)

As requested we have edited this paragraph and discussed the outcome of the recent *in vivo* results of two putative aminoacyltransferase-like TEII in an NRPS pathway in the recent ACIE paper (ACIE, 2021, doi.org/10.1002/anie.202103872) and the comparison with our *in vitro* outcomes. We have also added the text related to another putative TEII in the biosynthesis of rothibins, specialised metabolites recently discovered from *Streptomyces scabies*.

can the authors comment on the differences of these C* domains for dehydration and which aa might be involved in this catalysis?

The main focus of this MS is to biochemically characterize this new aminoacyltransferase-like TEII, LgnA. During the process, we found that the presence of LgnA is essential to the formation of the key intermediate (IV-Dhb-thioester) catalysed by LgnD-C₂. We have changed the relevant context to briefly address these C* domains for dehydration. This group of C* domains contains a typical HHxxDG catalytic motif emblematic of conventional C domains, where the second H is proposed to catalyse the amide formation. This is distinct to NoCB-M5C, in the nocardicin biosynthesis, which contains a rather unique H(790)HHxxDG in its active site where H790 was shown to be responsible for the generation of the cryptic transient Dha species and H792 is proposed to catalyse the amide bond formation during the course of the β-lactam formation. Further crystallography studies could shed light on the residue(s) responsible for the dehydration in these C* domains, including LgnD-C₂ domain.

line 321: what could be a mechanism to ensure this ration in vivo? Can the authors comment on the BGC and possible promoters/regulatory mechanisms?

The minimum BGC only includes 4 genes. The three NRPS genes *lgnB-lgnD* are in the same operon. The space between *lgnA* and these three genes were predicted to contain another promoter for the *lgnA* gene. It indicates that the transcriptional level of *lgnA* is likely to be different from that of the other three genes. We also changed the relevant context in the revised MS.

We have also corrected all of the minor grammatic errors, typos etc highlighted by the reviewer.

Reviewer #3:

There were many mistakes (particularly in the SI but also in the main text and figures) that made following the experiments very challenging, and coupled with the lack of information in the captions, missing graph axes etc. I have to admit that I got lost on many occasions even after multiple readings. Given this, I relied heavily on the figures to try and follow the manuscript, however even here I was lost on many occasions – apologies if my comments below therefore miss an experiment that is already included.

We apologize that the format of the original manuscript may have presented difficulties in terms of following the overall narrative. We have made significant changes to the revised manuscript to aid reader clarity.

Citation of the recent ACIE paper showing type II TE transfer should be included in a revised version.

We have added the recent ACIE paper to the manuscript as requested (please see comments above).

Starting with figure 2, the caption does not describe what is actually in the reaction

As requested we have amended the legend of Figure 2 to improve reader clarity.

Where is intermediate 8 bound, is it T0 or T1?

With new experimental evidence, we know that IV-Thr can be formed on LgnB-T₀ and LgnD-T₁, due to the promiscuous condensation catalysed by LgnD-C₁ domain. LgnD-T₁-tethered IV-Thr is the correct intermediate while LgnB-T₀-tethered IV-Thr is the aberrant intermediate. We have incorporated this data into the revised manuscript to address the point raised here.

What does the control reaction look like without proline?

We thank the reviewer for this suggestion, and we have carried out this experiment. Incubation of LgnA (TE_{II}), LgnB and LgnD with IV-CoA, Sfp, CoA, Thr and ATP in the absence of Pro only allowed us to identify the intermediate IV-Thr and IV-Dhb units as observed in our LC-MS analysis. We have added the control reaction without proline in the supplementary information Figure 12.

What are the levels of the intermediates shown in panels B and C? These look very similar

The figures containing the LC-MS traces have the same scale. In order to address the point raised, we have amended Figure 2 and moved the MS data of the chemical captured intermediates in the assay of LgnB and LgnD without LgnA to Supplementary Fig 6.

shouldn't there be an accumulation of 9 (9a) in the sample without LgnA? I see this in SI Figure 15, but these traces do not have any indication of a Y-axis at all and these traces do not seem to correspond with the traces in Figure 2.

During the chemical capturing experiments, we conducted the same conditions in the assays with or without the addition of LgnA in triplicate. The figures presented here are in the same scale. We have added the Y-axis of ion intensity in LC-MS traces in this figure. We have also compared the intensity of IV-Thr-Pro adducts in both assays in the presence and absence of

LgnA, IV-Thr-Pro adduct is significantly larger in the assay without LgnA than the one with LgnA.

The experiment is also somewhat flawed, as the role of LgnA in hydrolysis of the “incorrect” intermediate cannot be made as this enzyme is also apparently required for its formation – this could also be the role of the main TE domain

Based on our additional experimental evidence, it is concluded that LgnA transfers Thr intermediate from LgnB-T₀ to LgnD-T₁ domains. Due to the nonselective condensation reaction, IV-Thr unit can be also formed on the LgnB-T₀ domain, leading accumulation of this aberrant intermediate. LgnA hydrolyses the aberrant intermediate (LgnB-T₀-tethered IV-Thr).

Also for SI Figure 16, I do not see how this statement can be made, given that these intermediates will apparently be loaded onto the TE domain - this could just be hydrolysis over time due to multiple cycles of catalysis leading to higher signal with the same percentage of loss per cycle.

Given the new evidence provided in the beginning of the responses, LgnA transfers Thr from LgnB-T₀ to LgnD-T₁ domain and indeed hydrolyses the aberrant (LgnB-T₀-tethered IV-Thr) resulting in the accumulation of **12** as observed.

Figure 5 is missing further controls - add IV-CoA and use C1 in its apo form to see if IV is added; then also separate the C1T1 construct with Thr (or load it) to test that.

In order to address this point we have generated the LgnD-C₁-T₁ variant (C₁-T₁S492A), where the key Ser residue of LgnD-T₁ domain was changed to Ala, abolishing its thiolation function. This variant is equivalent to the C₁ domain alone. Incubation of LgnB and LgnD-C₁-T₁S492A with Thr, IV-CoA, CoA and ATP allowed the accumulation of **8** in the chemical capturing experiment (Supplementary Figure 19).

A key experiment - showing the LgnA loaded with an intermediate - is curiously absent. It would be very nice to be able to show this and to show which species it is (ie just Thr or also IV-Thr).

As requested we have carried out this key experiment and the data has been added to the revised manuscript. In our intact MS analysis, we indeed observed the Thr loaded LgnA while the LgnAS77A variant is inactive.

this means S2A mutants for T1 (LgnD_S2A_T1) and T2 (LgnD_S2A_T2) within the LanD construct to be able to tell where different intermediates are during the reaction.

The main focus of this MS is to characterize LgnA as an aminoacyltransferase to shuttle Thr from LgnB-T₀ to LgnD-T₁ domains in the early stage of the NRPS biosynthesis. We have indeed generated the LgnD-T₁ mutant where the Ser residue was changed to Ala as suggested. However, it is not our focus in this manuscript to elucidate the late stage of the biosynthesis.

This would also be important in Figure 3 where the site of dehydration is not known. It also links to the experiments in figure 5, where the site of addition of the acyl group is unclear (it would be on T0 or T1). Use of a synthetic probe on the T2 Ala mutant would allow the formation of alkene to be probed.

To locate the site of the dehydration, we conducted a series of assays of LgnB and LgnD-C₁-T₁ or LgnD-C₁-T₁(S492A) variant or LgnD-C₁-T₁-C₂ or LgnD-C₁-T₁(S492A)-C₂ variant with or without LgnA. Interpretation of accumulated chemically captured adducts in the assays of LgnB and LgnD-C₁-T₁ or LgnD-C₁-T₁(S492A) variant allowed us to conclude that IV-Thr can be formed in the LgnB-T₀ domain with or without the addition of LgnA. No IV-Dhb is formed.

In the assay of LgnB and LgnD-C₁-T₁-C₂ in the presence of LgnA, we observed the accumulation of an IV-Dhb adduct, indicating that C₂ catalyses the dehydration. When LgnD-C₁-T₁(S492A)-C₂ was included in the assay of LgnB with or without LgnA, no IV-Dhb is formed although IV-Thr unit was present in the reaction mixture, indicating that the dehydration must occur on the T₁ domain. When LgnA was omitted, only the IV-Thr adduct but not IV-Dhb was found, suggesting that LgnA may facilitate aminoacyltransfer. Later we showed that the exact functions of LgnA are to shuttle a Thr unit from the T₀ to T₁ domain and to remove the aberrant intermediate (LgnB-T₀-tethered IV-Thr).

Several attempts at the chemoenzymatic synthesis of IV-Thr-pantetheine were carried out (please see the final page of this document). We were able to synthesize IV-Thr-COOH and pantetheine arm as indicated in the Supplementary information. However, the coupling between IV-Thr-COOH and pantetheine in our hands was found to be extremely challenging. We instead synthesized IV-Thr-SNAC. However, this synthetic SNAC is not the substrate of LgnA.

We then adapted an *in situ* constitution of T₀ tethered IV-Thr experiment. We first incubated LgnB and LgnD-C₁-T₁ with Thr, IV-CoA, CoA, Sfp and ATP first, followed by ultra-centrifugal filtration of small molecules, such as Thr, IV-CoA, CoA and ATP. This ensured that the T₀ domain was fully loaded with an IV-Thr unit as shown in Fig 4iii. LgnA was then added to the remaining protein mixture. Intact MS analysis indicated that only holo- LgnB-T₀ and LgnD-T₁ domains were found and no IV-Thr was bound to LgnD-T₁ domain, strongly indicating that IV-Thr was removed from LgnB-T₀.

Mutation of the C1 and C2 active sites are also important controls to be able to show the involvement of each of these domains in the biosynthesis process, which is key given the important probably role of the type II TE in aminoacyl transfer.

We appreciate the suggestion. The focus of the MS is the key role of LgnA in the early stage of the biosynthesis. C₁ is the non-selective C domain, catalysing an IV-CoA and a Thr unit on the LgnB-T₀ or LgnD-T₁ domains as indicated in our chemical capturing and intact MS analysis. As indicated above, our chemical capturing experiment proved that LgnD-C₂ is responsible for the dehydration reaction. LgnD-C₂ is an unusual C-like domain that has the dehydration function. Without any structural information, it is difficult to predict which residue(s) is responsible for the dehydration reaction. It was worth noting that this group of C* domains is very similar to normal C domains that contain the typical motif, HHxxDG.

this should be extended to the LgnD-LgnB and LgnA-LgnB systems as well to understand these interactions as well

As requested in the revised manuscript we show that there are not interactions of LgnB ->LgnD-C₁T₁. Our microscale thermophoresis assay indicated the strong interaction of LgnA->LgnB.

the expts in SI Fig 18 appear to show not interaction, but this is likely due to their lack of loading - otherwise how can this TE domain transfer the units between T domains?

LgnA interacts with LgnD-T₁ domain as evidenced in our ITC analysis. No interaction between LgnA and LgnB was observed in our ITC analysis. The microscale thermophoresis assay indicated that LgnA indeed interacts with LgnB, a similar observation was made in the recent report in ACIE (2021, doi.org/10.1002/anie.202103872) where two putative TE_{II} orthologues, WS5 and WS20, were found to interact with the upstream didomains, WS22 and WS23, respectively.

The investigation of the truncated LgnD constructs is interesting but confusing - what is the interaction of the isolated T1 domain with LgnA?

Our new ITC evidence indicates that LgnA interacts with the LgnD-T₁ domain. This data has been added to the revised manuscript.

Figure SI 21 is very important - did the assays in the paper all use these optimal conditions (i.e low LgnA) ratio? These types of experiments have precedent in the last few years for other NRPS systems, although I did not see these cited here.

We apologise if this was not made clear in the original manuscript but we did indeed use the optimal conditions throughout the studies reported here. We also cited the biochemical precedent to show that the low TE_{II} ratio of its NRPS partner could contribute the peptide extension efficiency. We have added additional text to the revised manuscript to clarify these points.

Attempts at the synthesis of IV-Thr(O^tBu)-Ppant

Extensive efforts were made in the attempt to synthesise IV-Thr(O^tBu)-Ppant however the desired compound was found to be inaccessible to our hands. Below is a summary of the synthetic efforts which were made in the pursuit of this compound.

The synthesis began well with the installation of the isovaleryl chain onto Thr(O^tBu)-OMe followed by saponification of the methyl ester to the free acid for coupling.

Following the synthesis of this key building block it was envisaged that the coupling of this to pantetheine using standard thioester bond forming conditions (previously reported by Moore *et al. Org. Lett.* 2015, 4452) would be a relatively straightforward manner towards the target compound. LCMS appeared to show the formation of the desired product however following attempts at column chromatography no product could be recovered.

Next, we tried the same reaction with the inclusion of HOBt to increase the reactivity of the amide coupling reagents. Again, the same problem occurred with no material being recovered following purification.

We then resynthesized a new batch of pantetheine in a four-step synthesis from D-pantothenic acid hemicalcium salt as previously reported by Moore *et al.* We then tried coupling this material with IV-Thr(O^tBu)-OH. Again, LCMS pointed towards the formation of product however following extensive efforts at purification no material was recovered. At this point we believed that the product was unstable towards silica-based purification. In an effort to circumvent this problem we attempted to wash out the amide coupling reagent and then use the crude material in the next step.

LCMS of the crude material pointed towards the formation of the intermediate which was then treated with AcOH:H₂O to perform acetal deprotection. LCMS of this crude reaction mixture pointed towards successful product generation. Column purification of this material was then attempted but no material could be recovered. The whole synthesis was then repeated and reverse phase HPLC was attempted at the final step for product purification but again no product with the correct mass could be recovered. At this point the generation of IV-Thr(O^tBu)-Ppant through this particular route appeared to not be feasible. We conducted over 60 reactions over 2 months in our pursuit of this compound however we concluded that without significant extra time and a complete change in our synthetic approach the compound was not feasible to synthesise.

As a compromise we then turned our attention to the synthesis of IV-Thr(O^tBu)-SNAC. Taking the IV-Thr(O^tBu)-OH we had previously synthesised we were able to couple it to *N*-acetylcysteamine directly to give the desired product which was completely stable to column purification. The fact that we were able to readily access the SNAC appended compound points to the possibility that the pentathenine leads to significant instability in the thioester product.

REVIEWERS' COMMENTS

Reviewer #1 (Remarks to the Author):

The original manuscript was unclear for me to follow and I note reviewer number 3 also had difficulty. The authors have done an excellent job of rewriting the manuscript and addressing issues raised by myself and the other reviewers. There are many different parts to this paper, but I found it easy to follow and see how the results fit together.

Reference number 39 (<https://doi.org/10.1002/anie.202103872>), which the authors summarise in the discussion, appears to be the closest paper. This reference is very recently published in the high-impact journal *Angewandte Chemie*, and provides evidence of a TEII domain acting similarly in a different NRPS pathway. The current manuscript provides a second example of such a TEII domain. It also goes into deeper detail showing more of the mechanism and greatly adds to reference 39. This is valuable in supporting the existence of the transfer role and increasing our understanding. Accordingly, I think the current manuscript is suitable for publication in *Nature Communications*.

Reviewer #2 (Remarks to the Author):

The paper has been optimised a lot and I am happy with all responses and the additional experiments.

My only open question is regarding the protein ratios of LgnA vs LgnBD:

- although I appreciate Fig. 27, I find it too small and it can be easily enlarged.
- Moreover, would it be possible to quantify the amount of the proteins directly in the producer or use transcriptomics/RNAseq for that or is beyond the focus of the manuscript?

Reviewer #3 (Remarks to the Author):

The manuscript resubmitted here is a huge improvement on the previous version, and is not only much easier to follow but is also nicely fills in some gaps that had eluded characterization in the previous version. I'd like to congratulate the authors on making such an impressive effort to tackle the comments from the previous revision.

I only have a few minor points:

SI Fig 6 - can you really have a negative error?

SIG Fig 25 - is this really an "aberrant" intermediate as it is one involved in the biosynthesis?

I think it is difficult to say for sure that the dehydration occurs prior to condensation - I think the use of "suggests" here is a good way to express this.

Formation of 10 (bottom of page 7) - what is the identity of the "NRPS-tethered IV-Thr unit" - I'd suggest repeating the assay with LgnD-C1-T1-(S492A)-C2 to determine if this is T0 or T1. Also links to work on the bottom of page 9.

Page 8 - "chosed" should be "chose"

One point that I think is of interest is why the IV-Thr unit is cleaved but not Thr - if the modification of the amine is responsible, the TE could well be able to transfer AAs generally, but not acylated AAs. This will be very interesting to explore in future.

End of exptal (page 15) - why should this interaction not be visible by ITC? Some discussion here could be helpful.

Page 16 - spelling "(E)-2,3-dehydrotyrosin " is missing the "e"

Authors' responses to the reviewers:

We would like to thank the reviewers for their suggestions and comments. We have also included a point-by-point response to all of the comments raised by each reviewer.

Reviewer 1.

No responses are required

Reviewer 2.

Although I appreciate Fig. 27, I find it too small and it can be easily enlarged.

We are grateful for this suggestion, and have supplied the enlarged Fig. 27 in the revised manuscript.

Moreover, would it be possible to quantify the amount of the proteins directly in the producer or use transcriptomics/RNAseq for that or is beyond the focus of the manuscript?

This is an excellent point. These experiments will form a key part of our future research on this peculiar pathway. However, this experiment is beyond the scope of this study, which focuses on the biochemical characterization of a new aminoacyltransferase-like type II thioesterase (TE_{II}), LgnA.

Reviewer 3.

SI Fig 6 - can you really have a negative error?

This was a typographical error. We have changed SI Fig. 6 accordingly.

SIG Fig 25 - is this really an "aberrant" intermediate as it is one involved in the biosynthesis?

We agree with the reviewers point here, and have changed the context to 'shunt' product 12.

I think it is difficult to say for sure that the dehydration occurs prior to condensation - I think the use of "suggests" here is a good way to express this.

Agreed - we have changed the relevant text to reflect this.

Formation of 10 (bottom of page 7) - what is the identity of the "NRPS-tethered IV-Thr unit" - I'd suggest repeating the assay with LgnD-C1-T1-(S492A)-C2 to determine if this is T0 or T1. Also links to work on the bottom of page 9.

We appreciate the above suggestion, and have added a brief suggestion for NRPS-tethered IV-Thr unit (presumably T₀-tethered IV-Thr, *vide infra*). It is worth noting that we have already reported our results of the assay with LgnD-C1-T1-(S492A)-C2 in our revised manuscript that IV-Thr unit can be formed in T0 domain as the aberrant intermediate.

Page 8 - "chosed" should be "chose"

Many thanks for spotting this typographical error – now corrected.

One point that I think is of interest is why the IV-Thr unit is cleaved but not Thr - if the modification of the amine is responsible, the TE could well be able to transfer AAs generally, but not acylated AAs. This will be very interesting to explore in future.

Absolutely. Experiments to elucidate the molecular basis for this apparent 'hydrolytic selectivity' between L-Thr and IV-Thr will form a central part of our future work on this unusual NRPS cassette.

End of exptal (page 15) - why should this interaction not be visible by ITC? Some discussion here could be helpful.

Indeed, this is an interesting observation. We suspect that the interaction between LgnA and LgnB is thermodynamically distinct from the other interactions measured by ITC, which might be indicative of higher order conformational changes – we have added a sentence in the text to reflect this.

It is worth noting that the interactions between the WS22/WS23 (equivalent of LgnB) and WS5/WS20 (equivalent of LgnA) from the WS9326A pathway were characterised using MST, which might suggest difficulties in obtaining a K_d using the more 'traditional' ITC approach.

Page 16 - spelling "(E)-2,3-dehydrotyrosin " is missing the "e"

Many thanks for spotting this typographical error – now corrected.